# Effect of topographic slope on the export of nitrate in humid catchments: a 3D model study

Jie Yang[1], Qiaoyu Wang[1], Ingo Heidbüchel[2, 4], Chunhui Lu[1], Yueqing Xie[3], Andreas Musolff[2], and Jan H. Fleckenstein[2, 4]

[1]State Key Laboratory of Hydrology-Water Resources and Hydraulic Engineering, Hohai University, Nanjing, China
[2]UFZ - Helmholtz-Centre for Environmental Research GmbH, Department of Hydrogeology, Leipzig, Germany
[3]School of Earth Sciences and Engineering, University of Nanjing, Nanjing, China
[4]Hydrologic Modeling Unit, Bayreuth Center of Ecology and Environmental Research (BayCEER), University of Bayreuth, Bayreuth, Germany

*Correspondence to*: Chunhui Lu (clu@hhu.edu.cn)

**Key Points**

- Young water fractions of Q and ET are correlated to topographic slope negatively and positively, respectively

- Flatter landscapes tend to retain more nitrogen mass in the soil and export less nitrogen mass to the stream

- A large young streamflow fraction is not sufficient for high in-stream nitrate concentrations.

**Abstract.** Excess export of nitrate to streams affects ecosystem structure and functions and has been an environmental issue attracting world-wide attention. The dynamics of catchment-scale solute export from diffuse nitrogen sources can be explained by the changes of dominant flow paths, as solute attenuation (including the degradation of nitrate) is linked to the age composition of outflow. Previous data driven studies suggested that catchment topographic slope has strong impacts on the age composition of streamflow and consequently on in-stream solute concentrations. However, the impacts have not been systematically assessed in terms of solute mass fluxes and solute concentration levels, particularly in humid catchments with strong seasonality in meteorological forcing. To fill this gap, we modeled the groundwater flow and nitrate transport for a small agricultural catchment in Central Germany. We used the fully coupled surface and subsurface numerical simulator HydroGeoSphere (HGS) to model groundwater and overland flow as well as nitrate transport. We computed the water ages using numerical tracer experiments. To represent various topographic slopes, we additionally simulated ten synthetic catchments generated by modifying the topographic slope from the real-world scenario. Results suggest a negative correlation between the young streamflow fraction and the topographic slope. This correlation is more pronounced in flat landscapes with slopes < 1:60. Flatter landscapes tend to retain more N mass in the soil (including mass degraded in soil) and export less N mass to the stream, due to reduced leaching and increased degradation. The mean in-stream nitrate concentration shows a decreasing trend in response to

a decreasing topographic slope, suggesting that a large young streamflow fraction is not sufficient for high in-stream
concentrations. Our results improve the understanding of nitrate export in response to topographic slope in a temperate
humid climate, with important implications for the management of stream water quality.

**Keywords**: topographic slope, coupled surface-subsurface model, young streamflow, in-stream nitrate,
HydroGeoSphere

## 1 Introduction

Globally nearly 40% of land is used for agricultural activities [*Foley et al.,* 2005], which constitutes the major source
of pollution with nutrients such as nitrate (referred as to $N\text{-}NO_3$ in this study). Excess export of nitrate to streams
threatens ecosystem structure and functions, as well as human health via drinking water [*Vitousek et al.,* 2009; *Alvarez-*
*Cobelas et al.,* 2008; *Dupas et al.*, 2017]. This has been an environmental issue attracting attention in Germany and
world-wide. The dynamics of nitrate export from diffuse nitrogen (N) sources are regulated by the dominant flow
paths that determine the speed at which precipitation travels through catchments before it reaches the stream [*Jasechko*
et al. 2016]. The process is subject to both hydrological and biogeochemical influences mediated by various factors
(e.g. catchment topography, aquifer properties, redox boundaries). From the perspective of sustainable intensification,
process understanding and assessment of potential effects of catchment topography on nitrate export are critical for
the management of water quality in connection with agricultural activity.
Field observations in central German catchments indicate that in-stream nitrate concentrations ($C_Q$) show significant
differences in mean concentrations and seasonal variations between downstream areas with gentle topography and
more mountainous upstream areas [*Dupas et al.*, 2017; *Nguyen et al.,* 2022]. This provides strong evidence that
catchment topographic slope can influence the nitrate export. In terms of water age analyses, *Jasechko* et al. [2016]
using oxygen isotope data from 254 watersheds worldwide showed significant negative correlation between the young
(age < 3 months) streamflow fraction and the mean topographic gradient. They stated that young streamflow is more
prevalent in flatter catchments as these catchments are characterized by shallow lateral flow, while it is less prevalent
in steeper mountainous catchments as these catchments promote deep vertical infiltration. This statistically significant
trend is consistent with the common finding that fast shallow flow paths produce young discharge and potentially
influence the in-stream solute concentrations [*Böhlke* et al. 2007; *Benettin* et al. 2015; *Hrachowitz* et al. 2016; *Blaen*
*et al.* 2017]. However, apart from these data-driven analyses, a more mechanistic examination/explanation with the
aid of fully resolved flow paths is still required. Wilusz et al. [2017] used a coupled rainfall-runoff and transit time
model to investigate the young streamflow fraction, with a focus on the effect of rainfall variability rather than on
topography and solute export. Zarlenga et al. [2022] numerically quantified the relative contributions of hillslopes and
the drainage network to age dynamics in streamflow, considering the influences of transmissivity and recharge,
without focusing on topographic slope. The effect of topographic slope on $C_Q$ has rarely been subject to systematical
testing.
Seasonal fluctuation of $C_Q$ is commonplace in catchments under seasonal hydrodynamic forcing. Field observations
in mountainous central German catchments indicate that nitrate concentrations, as well as the mass load, in streams
vary seasonally, with maxima during the wet winter and minima during the dry summer [*Dupas et al.*, 2017]. Data-
driven analyses by *Musolff et al.* [2015] and *Dupas et al.* [2017] suggested the systematic seasonal (de)activation of
N source zones as an explanation for such seasonal variability. Under wetter winter conditions the near-surface N
source zones in agricultural soils are connected to the stream by fast shallow flow paths. Under drier summer
conditions those N source zones are deactivated because their direct hydrologic connectivity to the stream is replaced
with deeper flow paths [*Dupas et al.*, 2017]. Based on high-frequency monitoring in the Wood Brook catchment in
the UK, *Blaen et al.* [2017] also reported mobilization of nitrate from the uppermost soil layers during high flow
conditions via shallow preferential flow paths, which would not occur during base flow in drier periods. This behavior
leads to a seasonally-variable nitrate loading due to changing flow paths and the associated variation in transit time
that has been observed in many catchments [*Benettin et al.*, 2015; *Hrachowitz et al.*, 2016; *Kaandorp et al.,* 2018;
*Rodriguez et al.*, 2018; *Yang et al.* 2018]. However, how this fluctuation behaves in response to catchment land surface
topography has not been assessed systematically yet. Such an assessment could improve our understanding of nitrate
export from catchments of different topographic slopes not only in terms of the mean concentration but also regarding
its temporal variation patterns.
Given that most of the above studies used data driven analysis, numerical modeling is an effective tool for the analysis
of water flow, age and solute transport, eliminating the need for large amounts of field data. *Zarlenga and Fiori* [2020]
presented a physically-based framework to model transient water ages at the hillslope scale, which was later used to
investigate the different impacts of hillslopes and the channel network on water ages in catchments [*Zarlenga et al.*,
2022]. A number studies focused on numerically simulating the nitrogen fluxes (or loads) in soil and groundwater
[*Smith et al.,* 2004; *Rivett et al.,* 2008; *Lindström et al.,* 2010*; van der Velde et al.,* 2012; *Van Meter et al.,* 2017; *X.*
*Yang et al.,* 2018, 2019; *Kolbe et al.,* 2019; *Knoll et al.,* 2020; *Nguyen et al.,* 2021, 2022]. For example, *van der Velde*
*et al.* [2012] constructed a lumped numerical nitrate transport model for the Hupsel Brook catchment in the
Netherlands. *Lindström et al.* [2010] developed HYPE water quality model allowing for simulating the nitrogen fluxes
in soil. *Van Meter et al.* [2017] investigated the two-centuries nitrogen dynamics in the Mississippi and Susquehanna
River Basins using a TTD (transient time distribution) based transport approach. *X. Yang et al.* [2018] developed the
coupled mHM-Nitrate model, which can provide valuable insights int the spatial variability of water and nitrate fluxes
in catchment scale. *Nguyen et al.* [2021] further updated that model to the mHM-SAS model by implementing the
SAS-function based solute transport module [*Harman*, 2015, 2019; *Rinaldo et al.*, 2015; v*an der Velde et a*l., 2012],
allowing for simulating the nitrate export from a Mesoscale Catchment. However, most of these works provided little
information on the spatially-explicit details (such as the flow field) for interpreting the nitrate dynamics. Physically-
based hydrogeological models (like, e.g., HydroGeoSphere [*Therrien* et al., 2010]) resolve the spatially-explicit details
within a catchment including the full variability of 3D flow paths in the subsurface, helping to understand the
seasonally changing flow patterns in response to different catchment topographies. Additionally, the widely used fully-
coupled surface-subsurface technology simulates the catchment as an integrated system, providing details of surface
water-groundwater exchanges fluxes. These details help to identify paths of rapid discharge to the land surface that
can considerably improve the interpretation of nitrate-export patterns.
Transit time distributions (TTDs) have been widely used to interpret hydrological and chemical responses in catchment
outfluxes – both in discharge (Q) and in evapotranspiration (ET) [*Botter et al.*, 2010, 2011; *van der Velde et al.*, 2012;
*Heidbüchel et al.*, 2012; *Rinaldo et al.* 2015; *Harman et al.*, 2015; 2019]. They characterize how a catchment stores,
mixes and releases water as well as dissolved solutes at large spatial and temporal scales [*Benettin et al.*, 2015; *Harman*,
2015; *van der Velde et al.*, 2010, 2012; *Hrachowitz et al.*, 2015; *Van Meter et al.*, 2017]. Given that the nitrate
attenuation is linked to the age composition of outflow, the TTDs are ideal tools for interpreting the concentration
dynamics with regard to catchment topographic slope. Estimating water ages in natural catchments is still a challenge
due to varying climate conditions, as well as the errors in algorithms (e.g. errors in the flow field during particle
tracking) and limited computational capacity. *Yang et al.* [2018] used particle tracking to compute the age distributions
in the subsurface of a study catchment (while omitting the 4% of total discharge produced by direct surface runoff and
ignoring the frequent exchange fluxes that may be important for solute export due to their short transit times). Zarlenga
*et al.*, [2022] used a physically-based semi-analytical model to compute the transient water ages in a catchment,
however, without considering surface runoff and hydrological losses (e.g. ET). In this study we determined the age
compositions of Q and ET using numerical tracer experiments, where advective-dispersive transport of the tracers was
solved using the fully-coupled surface-subsurface framework of HydroGeoSphere. The computed age dynamics based
on the tracer concentrations were representative as the tracers were able to track all the flow processes such as surface
runoff, groundwater flow and surface-subsurface interaction.
In this study, we attempted to systematically assess the effect of catchment topographic slope on the nitrate export
dynamics in terms of mass fluxes, concentration levels and its seasonal variability. We also seek mechanical
explanations for the previously found behaviors from data-driven studies (like, e.g., *Jasechko et al.* [2016]) with the
help of fully resolved flow paths. First, we selected a real-world small agricultural catchment 'Schäfertal' in Central
Germany, which is characterized by strong seasonality in hydrodynamic forcing with associated shifts in the dominant
flow paths [*Yang et al.*, 2018]. This catchment is typical for many catchments with hilly topography under a temperate
humid climate. We created eleven model scenarios by adjusting the mean slope of the real-world catchment while
preserving the aquifer heterogeneity. Next, we modeled the water flow and nitrate transport for each catchment. The
flow and transport were solved using the fully coupled surface and subsurface numerical simulator HydroGeoSphere,
and the water ages were computed using numerical tracer experiments. Finally, the modeled flowpaths, water ages, N
mass fluxes and nitrate concentrations under various topographic slopes were analyzed. Through this study, we aimed
to (1) examine the relationship between topographic slope and N mass fluxes, and to (2) assess $C_Q$ and its seasonal
variation in response to different topographic slopes.

## 2 Data collection

### 2.1 Real-world and synthetic catchments

Our study was conducted on the catchment 'Schäfertal', situated in the lower part of the Harz Mountains, Central Germany (Figure 1a). The catchment has an area of 1.44 km$^2$. The hillslopes are mostly used for intensive agriculture while the valley bottom contains riparian zones with pasture and a small stream draining the water out of the catchment. The gauging station at the outlet of the catchment provides Q records. This gauging station is the only outlet for discharging water from the catchment, because a subsurface wall was erected underneath the gauging station across the valley to block subsurface flow out of the catchment. A meteorological station 200 m from the catchment outlet provides records of precipitation (J), air and soil temperatures, radiation and wind speed. The modeled catchment has a mean topographic slope of ~1:20, estimated using a cross-section perpendicular to the stream (Figure 1a). The aquifer thickness varies from ~5 m near the valley bottom to ~2 m at the top of the hillslope. Groundwater storage is low (~500 mm) in such a thin aquifer and mostly limited to the vicinity of the channel with the upper part of the hillslopes generally unsaturated. The stream bed has a depth of 1.5 m below the land surface. Aquifer properties (e.g. hydraulic conductivity) change from the hillslope, dominated by Luvisols and Cambisols, to the valley bottom, dominated by Gleysols and Luvisols [*Anis and Rode*, 2015]. Apart from that, the aquifer generally consists of two layers: the top layer of approximately 0.5 m thickness with higher porosity and a developed root zone from crops, and the base layer with smaller porosity due to high loam content [*Yang et al.*, 2018]. Subsequently, ten property zones were used (Figure 1b), with zonal parameter values following the model in *Yang et al.*, [2018] listed in Table 1.

Based on this real-world catchment, ten synthetic catchments were generated by adjusting elevations (land surface and aquifer bottom), such that the mean topographic slope ranges from 1:20 (steep) to 1:22, 1:25, 1:30, 1:40, 1:60, 1:80, 1:100, 1:200, 1:500 and 1:1000 (flat, Figure 1b). The aquifer depth and heterogeneity were preserved during the adjustments. In total, eleven catchments were used for flow and transport simulations. The catchment with the original topography (1:20) is selected as the base scenario.

### 2.2 Climate

The considered climate for the catchments was derived from the catchment 'Schäfertal' located in a region with temperate humid climate and pronounced seasonality. According to the meteorological data records from 1997 to 2007, the mean annual J and Q (per unit area) are 610 mm and 160 mm, respectively. Actual mean annual ET based on the ten-year water balance ($J = ET + Q$) is 450 mm. Mean annual potential ET is 630 mm [*Yang et al.*, 2018]. The humid climate is representative for wet regions, quantified by an aridity index (J / potential ET [*Li et al.,* 2019]) of 1.0. The ET is the main driver of the hydrologic seasonality as the precipitation is more uniformly distributed across the year (Figure 1c).

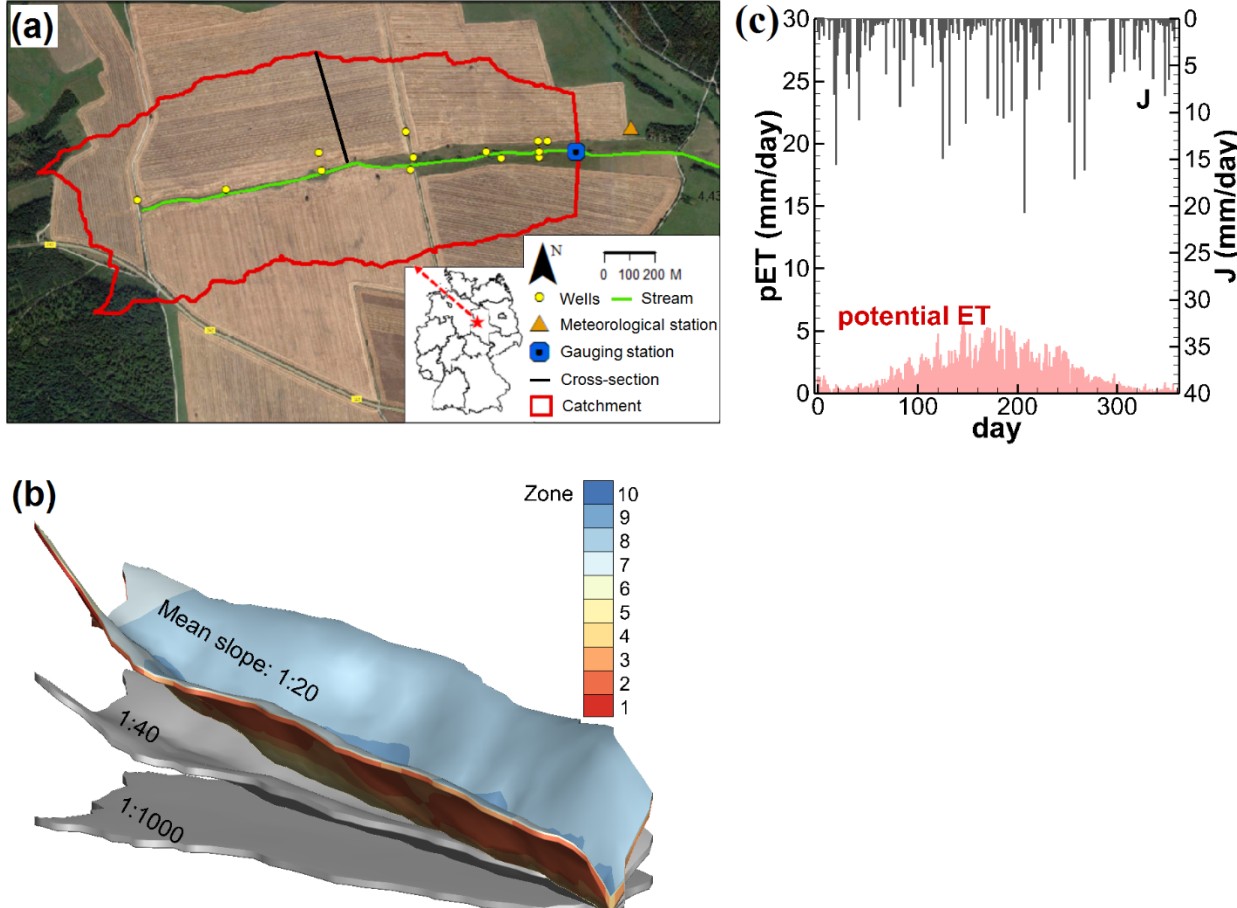

**Figure 1.** (**a**) The catchment 'Schäfertal', Central Germany (background image from © Google Maps). (**b**) The catchments with mean topographic slopes of 1:20, 1:40 and 1:1000. (**c**) The measured precipitation *J* and the estimated potential evapotranspiration *ET* for the year 2005 under the humid climate [*Yang et al.*, 2018]. Ten aquifer property zones in (b) were defined in the subsurface of the catchment for zonal parameter values (e.g. hydraulic conductivity).

## 3 Methods

### 3.1 Flow and nitrate transport

*Flow model*

It is necessary to solve both groundwater and surface water flow because the spatially-explicit details in the catchment including the specific flow paths and exchange fluxes are necessary to interpret the effect of varying topographic slope on nitrate transport. We simulated the flow system using the fully coupled surface and subsurface numerical model HydroGeoSphere, which solves for variably saturated groundwater flow with the Richards' equation and for surface flow with the diffusion-wave approximation of the Saint-Venant equations [*Therrien* et al., 2010]. Additionally, the exchange flux between groundwater and surface water can be implicitly simulated. The nitrate transport is simulated

in the groundwater flow, surface flow and exchanges fluxes by solving the advection-dispersion-diffusion equation
describing the conservation of nitrate mass. HydroGeoSphere has been successfully used to simulate catchment
hydrological processes and solute transport in many studies [e.g. *Therrien et al.*, 2010; *Yang et al.*, 2018], therefore
governing equations and technical details are not explicitly repeated here.
In our previous work *Yang et al.* [2018], a hydrological flow model has already been established for the catchment
'Schäfertal'. It was calibrated against measured groundwater levels and stream discharge Q. The optimized parameter
values are listed in Table 1. In this work, we performed our simulations based on that flow model, with the nitrate
transport process being added while maintaining the model setup. We provide a brief review of that flow model here.
Readers may refer to *Yang et al.* [2018] for a full description of the model and its calibration.
The modeled subsurface of the catchments was discretized into 9 horizontal layers between the land surface and the
aquifer base, with thinner layers in the upper part (0.1 m) to better represent the unsaturated zone and compute the ET.
In total, the subsurface was discretized by a mesh of 13860 prisms, with the horizontal size of the prisms ranging from
30 to 50 m. The topmost 1540 triangles were used to discretize the surface domain, where surface flow was simulated.
Ten property zones for the subsurface were defined (Figure 1b), being assigned with the zonal hydraulic conductivity
and porosity values (Table 1). ET was simulated as a combination of plant transpiration from the root zone (top 0.5 m
soil) and evaporation down to the evaporation depth (0.5 m), which are both constrained by soil water saturation.
Regarding the flow boundary conditions, spatially uniform and temporally variable J was applied to the land surface.
Spatially constant and temporally variable potential ET was applied to the aquifer top to calculate the actual ET. The
bottom of the aquifer was considered an impermeable boundary. A critical depth boundary condition was assigned to
the catchment outlet to simulate the stream discharge Q, which was compared to the measured Q during the calibration.
The software PEST [Doherty and Hunt, 2010] was used for the transient calibration. After calibration, the time-
variable groundwater levels were well replicated by the flow model for most of the wells, with mean coefficients of
determination ($R^2$) of 0.43. The fit between the simulated and measured Q was satisfactory with a $R^2$ of 0.61. The
calibrated model successfully simulated the flow system from 1997 to 2007.
In this study, we continued to use the above-described model setup, including the mesh, the parameters and the flow
boundary conditions, for the eleven catchments with different topography. Note that the mesh was adapted to the
change of the topography by changing node elevations vertically. However, to simplify the flow simulation and the
age computation (described in section 3.2), we selected the year 2005 as a representative year and assumed that all the
years have the identical climate (J and potential ET) as the year 2005. Therefore, J and potential ET of 2005 (Figure
1c) were cycled and applied to the catchments for all the simulated years.

**Table 1**. The key flow parameters and their values following *Yang et al.*, [2018].

| Parameter | Process | Type | Value |
|---|---|---|---|
| Hydraulic conductivity | Subsurface | zonal | Zonal values (range [$3.6 \cdot 10^{-5}$ - 2.0] m day$^{-1}$) |
| Porosity | Subsurface | zonal | Zonal values (range [0.01 - 0.35]) |
| Residual saturation | Subsurface | uniform | 0.08 [-] |
| Inverse of air entry pressure $\alpha$ | Subsurface | uniform | 3.6 m$^{-1}$ |
| Pore-size distribution index $\beta$ | Subsurface | uniform | 2 [-] |
| Manning roughness coefficient | Surface | uniform | $6.34 \cdot 10^{-6}$ day m$^{-1/3}$ |
| Longitudinal dispersivity | Transport | uniform | 8 m |
| Lateral and vertical dispersivity | Transport | uniform | 0.8 m |
| Molecular diffusion coefficient | Transport | uniform | $10^{-9}$ m$^2$ s$^{-1}$ |
| Degradation coefficient | Transport | uniform | 0.009 day$^{-1}$ |
| **Transpiration fitting parameters:** | | | |
| C1 | ET | uniform | 0.17 [-] |
| C2 | ET | uniform | 0.00 [-] |
| C3 | ET | uniform | 3.00 [-] |
| **Transpiration limiting saturations:** | | | |
| Wilting point | ET | uniform | 0.1 [-] |
| Field capacity | ET | uniform | 0.2 [-] |
| Oxic limit | ET | uniform | 0.9 [-] |
| Anoxic limit | ET | uniform | 1.0 [-] |
| **Evaporation limiting saturations:** | | | |
| Minimum | ET | uniform | 0.1 [-] |
| Maximum | ET | uniform | 0.2 [-] |




***Transport boundary conditions and parameters***
The nitrogen (N) pool is formed in the soil zone of the catchments, representing a nitrate source zone. The N pool is
controlled by various complex processes. It is replenished by external inputs from atmospheric deposition, biological
fixation, animal manure from the pasture area, and fertilizer from the farmland on the hillslopes. Nitrate that can be
transported with water is formed and leached from this N pool by a microbiological immobile-mobile exchange
process [*Musolff* et al., 2017; *Van Meter* et al., 2017]. In our study, we employed the simplified framework by *Yang*
*et al.*, [2021] to track the fate of N in the N pool (Figure 2a). This framework was derived from the ELEMeNT
approach (Exploration of Long-tErM Nutrient Trajectories, Van Meter et al., 2017), which uses a parsimonious
modeling framework to estimate the biogeochemical legacy nitrate loading in the N pool and the N fluxes leaching
from the N pool to the groundwater. This framework assumes that total N load in the N pool is comprised by inorganic
N (SIN) and organic N (SON). Two types of SON are distinguished: active organic N (SON$_a$) with faster reaction
kinetics and protected organic N (SON$_p$) with slower reaction kinetics. It is assumed that the external N input
contributes only to the SON. The SON is mineralized into SIN. The SIN is further consumed by plant uptake and

denitrification, and finally leaches to groundwater as dissolved inorganic N (DIN, representing mainly nitrate in the studied catchment [*Yang et al.*, 2018; *Nguyen et al.*, 2021]). The framework is acceptable due to the fact that most of the nitrate fluxes from source zones has undergone biogeochemical transformation in the organic N pool [*Haag and Kaupenjohann,* 2001]. The framework simplifies complexities of different N pools and transformations via mineralization, dissolution, and denitrification within the soil zone [Lindström et al., 2010], while preserving the main pathway for nitrate leachate.

The governing equations to calculate these N fluxes follow the ones in *Yang et al.*, [2021]. A specific portion (*h*) of the external N input contributes to the $SON_p$ pool, and the rest contributes to the $SON_a$ pool. The portion *h* is the land-use dependent protection coefficient [*Van Meter et al.*, 2017]. The mineralization and denitrification are described as first order processes with rate coefficients $k_a$, $k_p$, and $\lambda_s$ respectively, using:

$$MINE_a = k_a \cdot f(temp) \cdot SON_a \tag{1}$$

$$MINE_p = k_p \cdot f(temp) \cdot SON_p \tag{2}$$

$$DENI_s = \lambda_s \cdot SIN \tag{3}$$

where $MINE_a$, $MINE_p$, $DENI_s$ (kg ha$^{-1}$ day$^{-1}$) are the mineralization rates for $SON_a$ and $SON_p$, and denitrification rate for SIN. $k_a$, $k_p$, and $\lambda_s$ (day$^{-1}$) are coefficients for the first order processes. $f(temp)$ is a factor representing a constraint by soil temperature [*Lindström et al.*, 2010]. Note that the mineralization and plant uptake occur in the N pool. Denitrification can occur in both the N pool and later in groundwater. The plant uptake rate UPT follows the equation used in the HYPE model [*Lindström* et al., 2010]:

$$UPT = \min(UPT_P, 0.8 \cdot SIN) \tag{4}$$

$$UPT_P = p1/p3 \cdot \left(\frac{p1-p2}{p2}\right) \cdot e^{-(DNO-p4)/p3} / \left(1 + \left(\frac{p1-p2}{p2}\right) \cdot e^{-(DNO-p4)/p3}\right)^2 \tag{5}$$

where $UPT$ and $UPT_P$ (kg day$^{-1}$ ha$^{-1}$) are the actual and potential uptake rates. The computation of $UPT_P$ considers a logistic plant growth function. *DNO* is the day number. *p1, p2, p3* are three parameters depending on the crop/plant type, they are in the units of (kg ha$^{-1}$), (kg ha$^{-1}$), and (day), respectively. *p4* is the day number of the sowing date. The leaching process allows for SIN to leach from the soil (N pool) to the groundwater. The leaching rate *LEA* (kg ha$^{-1}$ day$^{-1}$) is defined as a first order process as:

$$LEA = f \cdot SIN / \Delta t \tag{6}$$

$$f = \left(1 - exp^{-a\frac{wal}{\theta d}}\right) \tag{7}$$

$$wal = q \cdot \Delta t \tag{8}$$

where *f* is a factor, ranging between [0, 1], to determine the portion of SIN that leaches into groundwater during a time step $\Delta t$. *a* is unit-less leaching factor. $\theta$ is the soil porosity. *d* is the soil depth. *wal* [L] is the water available for leaching during $\Delta t$. *wal* can be estimated using the Darcy fluxes *q* [LT$^{-1}$], which are provided by the flow simulations for each cell of the mesh. Physically, *f* is a function of the ratio between *wal* and the volume of soil voids $\theta \cdot d$, representing the ability of water to flush the SIN. This formulation of *LEA* is modified from the ones used in *Pierce et al.*, [1991], *Shaffer et al.* [1991] and *Wijayantiati et al.* [2017], to comply with the spatially-distributed HydroGeoSphere model.

**Table 2**. The parameters for the N pool and nitrate transport. The parameters with a range are calibrated. The
adjustable ranges are selected to cover the values that the parameters can potentially take on or the values reported
by the referred literature.

| Parameter | Description | Range | Reference | Best-fit value |
|---|---|---|---|---|
| N pool | | | | |
| $d$ | Soil depth | Fixed | *Yang et al.* [2018] | 0.5 m |
| *N Input* | N external input | Fixed | *Nguyen et al.* [2021] | 180 kg ha$^{-1}$ yr$^{-1}$ |
| $h$ | protection coefficient | Fixed | *Van Meter et al.* [2017] | 0.3 [-] |
| $k_a$ | Mineralization coef. (DON$_a$) | [0 - 0.7] | *Yang et al.* [2021] | 0.011 day$^{-1}$ |
| $k_p$ | Mineralization coef. (DON$_p$) | [0 - 0.7] | *Yang et al.* [2021] | 0.0008 day$^{-1}$ |
| $\lambda_s$ | Denitrification coef. (soil) | [0 - 0.7] | *Yang et al.* [2021] | 0.0007 day$^{-1}$ |
| $p1$ | Parameter for plants-uptake | [60 - 160] | *Van Meter et al.* [2017] | 160 kg ha$^{-1}$ |
| $p2$ | Parameter for plants-uptake | [0 - 10] | | 9.8 kg ha$^{-1}$ |
| $p3$ | Parameter for plants-uptake | [1 - 60] | | 25.6 day |
| $p4$ | Parameter for plants-uptake | Fixed | | 63 day |
| $a$ | Leaching factor | [0 – 100] | | 0.154 [-] |
| Transport | | | | |
| $\lambda$ | Denitrification coef. (water) | [0 - 0.7] | *Yang et al.* [2021] | 0.0072 day$^{-1}$ |
| $a_L$ | Longitudinal dispersity | Fixed | | 8 m |
| $a_T$ | Transverse dispersivity | Fixed | | 0.8 m |


The N pool is positioned on the top part of the aquifer, used as a boundary condition for the DIN (nitrate) transport.
Advective-dispersive transport of DIN in the flow system is simulated using HydroGeoSphere (Figure 2b).
Degradation (denitrification in groundwater) during transport is considered as a first order process. Degradation is not
considered on the land surface (denitrification in surface flow), where aerobic conditions likely deactivate
denitrification and residence time is short. To implement the evapoconcentration effect in the transport model, ET is
assumed to remove DIN mass without altering the DIN concentration of the water, and to inject that mass back to the
SIN pool. This represents a precipitation process from DIN to SIN, which is the inverse process of leaching (Figure
2b). There are two reasons for doing that: (i) the physical process of ET causing the immobilization of DIN can be
mathematically considered, and (ii) the N mass balance can be conserved as the plants-uptake is already considered
in the N pool according to the plant growth function (Equation 4 and 5), being independent from the ET flux.
Regarding the parameters, the soil depth, within which the N pool is implemented, is set to 0.5 m. N external input is
180 kg ha$^{-1}$ yr$^{-1}$ according to Nguyen et al. (2021), where the nitrate balance was simulated for the larger upper Selke
catchment that contained our studied catchment. The external N input is assumed to be spatiotemporally constant due
to the limited information on its variation in space and time. The protection coefficient *h* is fixed as 0.3 according to
the values reported in *Van Meter et al.* [2017]. The sowing date *p4* is fixed as 63 days according to the fact that sowing
activities and plant growth start in early March. Longitudinal and transverse dispersivity values were 8 m and 0.8 m,
respectively. Other parameters were set to be adjustable and calibrated (Table 2).

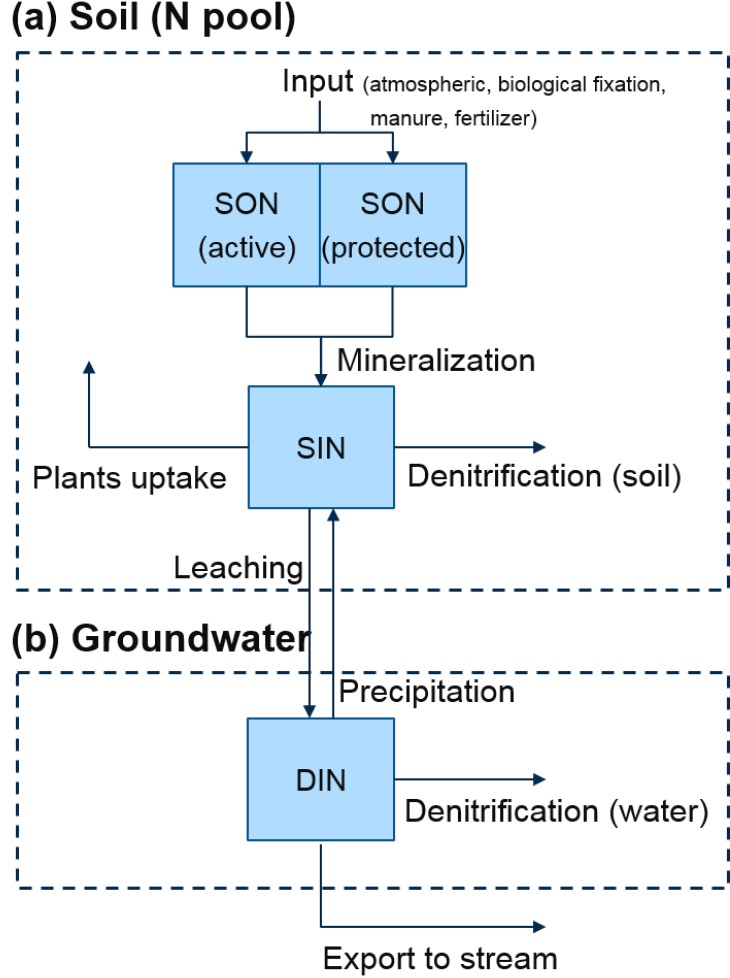


**Figure 2.** Conceptual framework for nitrogen (N) fluxes (a) in the soil (N pool), and (b) after leaching into the
groundwater.


*Transport calibration*
As the flow parameters (e.g., hydraulic conductivity and porosity) were already calibrated in *Yang et al.* [2018] using
data sets of discharge and groundwater levels. In this study, the calibration was only performed for the transport to get
reasonable parameter values for the N pool and N transport. The software package PEST [*Doherty and Hunt,* 2010]
was used. PEST uses the Marquardt method [*Marquardt*, 1963] to minimize a target function by varying the values
of a given set of parameters until the optimization criterion is reached. We used the measured $C_Q$ and N surplus as the
target variables for comparison with the simulated ones. The N surplus, which is the annual amount of N remaining
in the soil after consumption by plant-uptake, was estimated as 48.8 kg ha$^{-1}$ yr$^{-1}$ [*Yang et al.,* 2021]. As two different
data sets ($C_Q$ and N Surplus) were used, a weighting scheme was used such that the defined multi-objective function
was not dominated by one data set.
Note that the entire model calibration (for flow and transport) actually followed a procedure of two steps: first for
flow, and second for transport. Alternatively, the flow and transport parameters can be calibrated at one step by
defining the multi-objective function using all the data sets (discharge, groundwater levels, $C_Q$ and N surplus). The
potential effect of the two different calibration procedures on the modeling results should be further explored, however,
being out of the main focus of this study. We consider the two-step calibration procedure to be acceptable, because
our result showed that it was sufficient to reach an acceptable model performance for both flow and transport
(described later).
Several transport parameters were fixed at the values selected according to prior information, such that the degree of
freedom in the calibration can be reduced as much as possible (Table 2). In total eight parameters were adjustable and
calibrated, because they were the key parameters to determine the N fluxes in soil and groundwater. Their adjustable
ranges were selected according to the literature or to cover the values that the parameters can realistically reach (Table
2). The calibration was carried out for the period from Jul 1999 to Jul 2003, during which the data sets are available.
First, the flow and transport were simulated in the catchment of the base scenario (original topography, section 2.1).
Secondly, PEST was used to obtain a best fit between the simulated results and the data sets by varying the parameter
values. Note that the simulation period from Jul 1999 to Jul 2003 was only used for model calibration, rather than for
the actual simulations with the eleven catchments of different topographic slope. After calibration, the model with the
best-fit parameter values can well replicate the measured $C_Q$ with a Nash-Sutcliffe efficiency (NSE) of 0.75 (see Figure
S1 in the supporting information). The simulated N surplus was 50.7 kg ha$^{-1}$ yr$^{-1}$, comparable to the measured value.
The best-fit parameter values from the base scenario were used for all other scenarios with catchments of different
topographic slope, assuming that the parameters do not change with the change of topographic slope. In total, we
simulated the flow and nitrate transport for eleven scenarios (11 catchments of different topographic slope). For each
scenario, the simulations were run for 100 years with identical boundary conditions for each year. The first 99 years
were used as a spin-up phase to assure a dynamic equilibrium (i.e. to achieve simulated variables, such as heads and
concentrations, that are identical between years), and the last year was used for actual observation and analysis. The
CPU time of each simulation was ~4 hours.

## 3.2 Water ages

The water stored in a catchment (storage), Q and ET can all be characterized by age distributions, for they comprise
water parcels of different age from precipitation events that occurred in the past. The age distributions need to be
calculated for each aforementioned scenario to assess the responses of water ages on catchment topographic slope.
Our model setup (with virtual catchments and identical climate for each year) allowed us to perform long-term
numerical tracer experiments and to extract the age distributions.
We assumed that inert tracers of uniform concentration existed in precipitation. The tracers were applied to the land
surface as a third-type (Cauchy) boundary condition and were subjected to transport modeling. Tracer can exit the
aquifer via the outfluxes Q and ET. We considered a period of 200 years for the tracer experiments, which was
sufficiently long to ensure convergence of the computed water ages. The 200 years period was partitioned into 2400
months ($\Delta t$ = 1 month). A different tracer was used for each of the periods resulting in a total of 2400 distinct tracers.
The injection of tracer $i$ started with the precipitation at the beginning of its associated period $t_0^i$ and lasted throughout
the period. The advective-dispersive multi-solute transport was simulated using HydroGeoSphere. The first 199 years
of the simulation period were used as a spin-up phase to ensure a dynamic equilibrium of the calculated ages,
minimizing the influence of the initial conditions. The last year was used for the actual observations and the
computation of age distributions. Solving the transport of the 2400 tracers would be computationally expensive.
However, because the climate (flow boundary conditions) was identical for each year, the transport simulation was
performed only for the first 12 tracers that covered the course of a year. Based on these results, the results for the other
2388 tracers were manually reproduced (e.g., by shifting the concentration breakthrough curves of the 12 tracers in
time while maintaining the shapes).
For each tracer, the breakthrough curves of the mass-fluxes of Q and ET, as well as the mass in storage were reported.
For a specific time $t$, the age distributions for Q/ET/storage were computed by calculating the mass fraction of each
tracer using:

$$p_{Q/ET/S}(T,t) = \frac{M^i(t)}{\Delta t \sum M^i(t)} \tag{9}$$


where $p_Q(T,t)$, $p_{ET}(T,t)$ are the age distributions of Q, ET (equivalent to backward transit time distributions - TTDs),
and $p_S(T,t)$ is the age distributions of water in storage (equivalent to the residence time distribution - RTD). $M^i(t)$ is
the mass-flux of the tracer $i$ in Q or ET, or the mass stored in catchment at time $t$, $\sum M^i(t)$ is the sum of $M^i(t)$ over
all tracers. $T$ is the age ranging within $[t - t_0^i - \Delta t, t - t_0^i]$ for tracer $i$.
For each scenario, the CPU time of the tracer experiment was ~8 hours. Based on the age distributions, we calculated
the mean discharge age $T_Q(t)$, which is equivalent to the mean discharge transit time (simply referred to as 'discharge
age' in the following sections). We calculated the young water fraction in streamflow $YF_Q(t)$, which is the fraction of
streamflow with an age younger than three months (also referred to as 'young streamflow fraction' [*Jasechko et al*.
2016]). Similarly, the ET age $T_{ET}(t)$ and the young water fraction in ET $YF_{ET}(t)$ can be calculated as well (more
details are described in Text S1 of the supporting information). Their responses to changes in topographic slope were
analyzed.

**4 Results and discussion**
**4.1 Dynamics of water ages and nitrogen fluxes**
Driven by the seasonality of the climate, the simulated Q, the young water fractions *YF*, and the water ages all show
seasonal fluctuations. Figure 3 shows these fluctuations for the base scenario (original topography). Q reaches its
maximum towards the end of the wet winter in late February and reaches its minimum during the drier late summer
in mid-September. Total Q consists of a portion of groundwater discharge (including the flow via vadose zone) and a
portion generated via surface-runoff during events of high precipitation (Figure 3a). The calculated $YF_{ET}$ is smallest
in April and largest in November (Figure 3b), while $YF_Q$ is smallest in August and largest in February. ET generally
has larger young water fractions than Q as ET has a higher probability to remove young water from the shallow soil
rather than the older water from the deeper aquifer. Especially during the dry season (summer), most precipitation can
be quickly removed by ET. The water ages of Q and ET show generally opposite fluctuation patterns for $YF$ (Figure
3c). The ET age ranges from 70 to 115 days, being younger than Q that has the age ranging from 109 to 180 days.

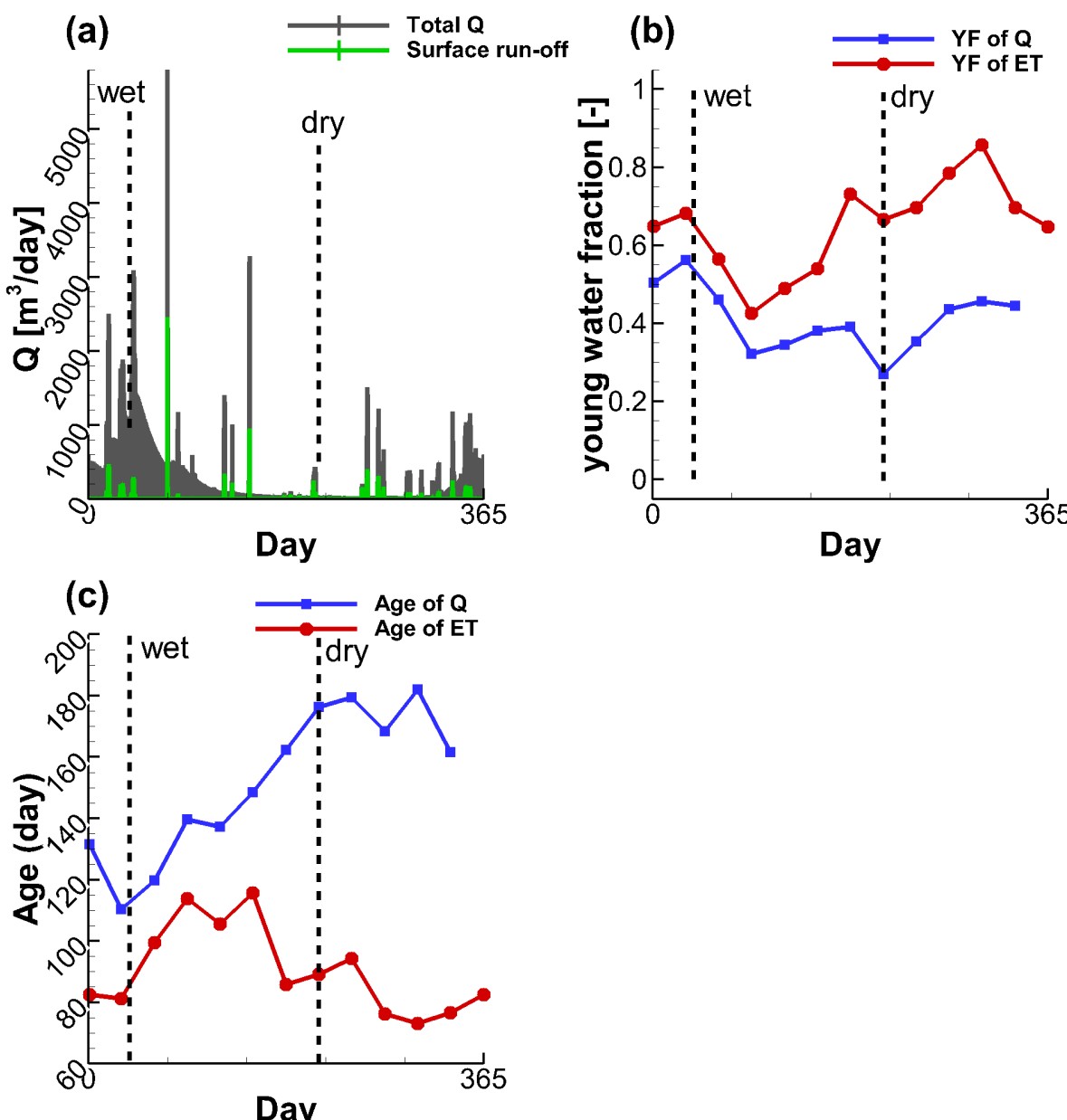


**Figure 3.** Simulated (**a**) Q, (**b**) young water fractions in streamflow ($YF_Q$) and evapotranspiration ($YF_{ET}$), and (**c**) water
ages for the catchment of the base scenario. The $YF$ and water ages are monthly averages.

The simulated $C_Q$ shows strong seasonality with maxima in the wet and minima in the dry period, fitting the measured $C_Q$ data well (Figure 4a). Figure 4b lists the calculated annual N mass balance in the catchment of the base scenario. The organic (SONa + SONp) and inorganic (SIN) N load in the soil are 470 kg ha$^{-1}$ and 43 kg ha$^{-1}$, respectively. The SON accounts for 92% of the total N load, which is consistent with the study of Stevenson [1995] where the organic N fraction was reported to be greater than 90%. The mineralization converts SON into SIN with a rate of 180 kg ha$^{-1}$ yr$^{-1}$. This rate is equal to the external N input because this way a steady-state of the annual N mass balance was reached in the simulations. About 76% of the input N flux is taken up by the vegetation (136 kg ha$^{-1}$ yr$^{-1}$). 20% is consumed by denitrification (36 kg ha$^{-1}$ yr$^{-1}$), either in the soil (before leaching) or in the groundwater (after leaching). The remaining 4% reaches the stream water and is exported out of the catchment (6 kg ha$^{-1}$ yr$^{-1}$). The simulated mineralization flux is within the range of [14–187] kg ha$^{-1}$ yr$^{-1}$ reported by *Heumann et al.* [2011] for their study sites in central Germany. The simulated plant uptake and leaching fluxes are comparable to the values suggested in Nguyen et al. [2021] for the same area (120 kg ha$^{-1}$ yr$^{-1}$ for plant uptake and [15–60] kg ha$^{-1}$ yr$^{-1}$ for leaching). The simulated denitrification rate is within the range [8–51] kg ha$^{-1}$ yr$^{-1}$ reported in Hofstra and Bouwman [2005] for 336 agricultural soils located worldwide. Moreover, 80% and 20% of the leaching N are consumed by denitrification during transport in the groundwater and exported to stream water, respectively. These portions are generally comparable to those reported in Nguyen et al. [2021] (61% and 39%, respectively). Therefore, the simulated N loads and fluxes for the catchment of the base scenario are considered to be acceptable.

Figure 4c shows the temporal variation of the N load and fluxes. It demonstrates that low levels of SIN are maintained by high plant-uptake before the dry summer arrives (May – June), such that there is little SIN available for leaching. The SIN load reaches its minimum when plant uptake reaches its maximum (marker a in Figure 4c). The cessation of plant-uptake during the dry period leads to the increase of the SIN load as well as the increase of the leaching rate. The mineralization in winter is significantly reduced due to the dropping temperatures, cutting the SIN supply. This results in the SIN load reaching its high peak in the middle of November (marker b in Figure 4c) and subsequent decrease due to increased leaching and eventually plant uptake. These seasonal fluctuation patterns are generally consistent with the knowledge of N fluxes reported in previous studies [Dupas et al., 2017; Nguyen et al., 2021]. For DIN load in water, it reaches its maximum generally when the leaching weakens in the beginning of March (marker c in Figure 4c), and reaches the minimum just before the leaching process becomes active again in the end of August (marker d in Figure 4c). These low and high peaks of SIN and DIN loads can also be identified by their spatial distributions in the catchment (see Figure S2 in the supporting information).

Seasonal variations of $C_Q$ can be directly influenced either by the fluctuation of the nitrate leaching into groundwater, or by fluctuations in the degradation in groundwater associated with varying transit times (quantified by the young water fraction in streamflow $YF_Q$). These two influences represent the effect from the variability in N source and in N transport, respectively. Linear regression analysis shows that $C_Q$ is correlated with leaching flux rate and $YF_Q$ with Spearman rank-correlation coefficients of 0.1 and 0.34, respectively (Figure 5). The seasonal fluctuations of $C_Q$ and leaching flux are temporally out of phase. The maximum leaching occurs in December, while the maximum $C_Q$ is

reached two months later in February (Figure 5a). The minimum leaching occurs in April, while the minimum $C_Q$ is
reached around September. This behavior indicates that $C_Q$ responds later to the changes in N leaching, which is
reasonable because the leaching nitrate needs time to travel from the shallow soil to streamflow. The fluctuation of
$C_Q$ and $YF_Q$ are more synchronized, proven by the fact that both maxima are reached in February (wet, Figure 5b) and
minima occur generally in the dry summer time. Field observations in mountainous central German catchments also
indicate that $C_Q$ varies seasonally, with maxima during the wet winter and minima during the dry summer [Dupas et
al., 2017]. These seasonal fluctuations of $C_Q$ and $YF_Q$ were frequently explained using the "inverse storage effect"
[*Harman*, 2015; *Yang et al.* 2018]: during the wet season Q has a strong preference for young water associated with
higher concentrations, which would not occur during dry periods due to the deactivation of the shallow fast flow
processes. These patterns generally suggest that the $C_Q$ fluctuation is more attributed to the variability in the N
transport rather than to the variability in the N source, echoing previous observations that 80% of the leaching N mass
is degraded during transport. However, it is still hard to tell whether the N source or the N transport is dominating the
$C_Q$ fluctuation.


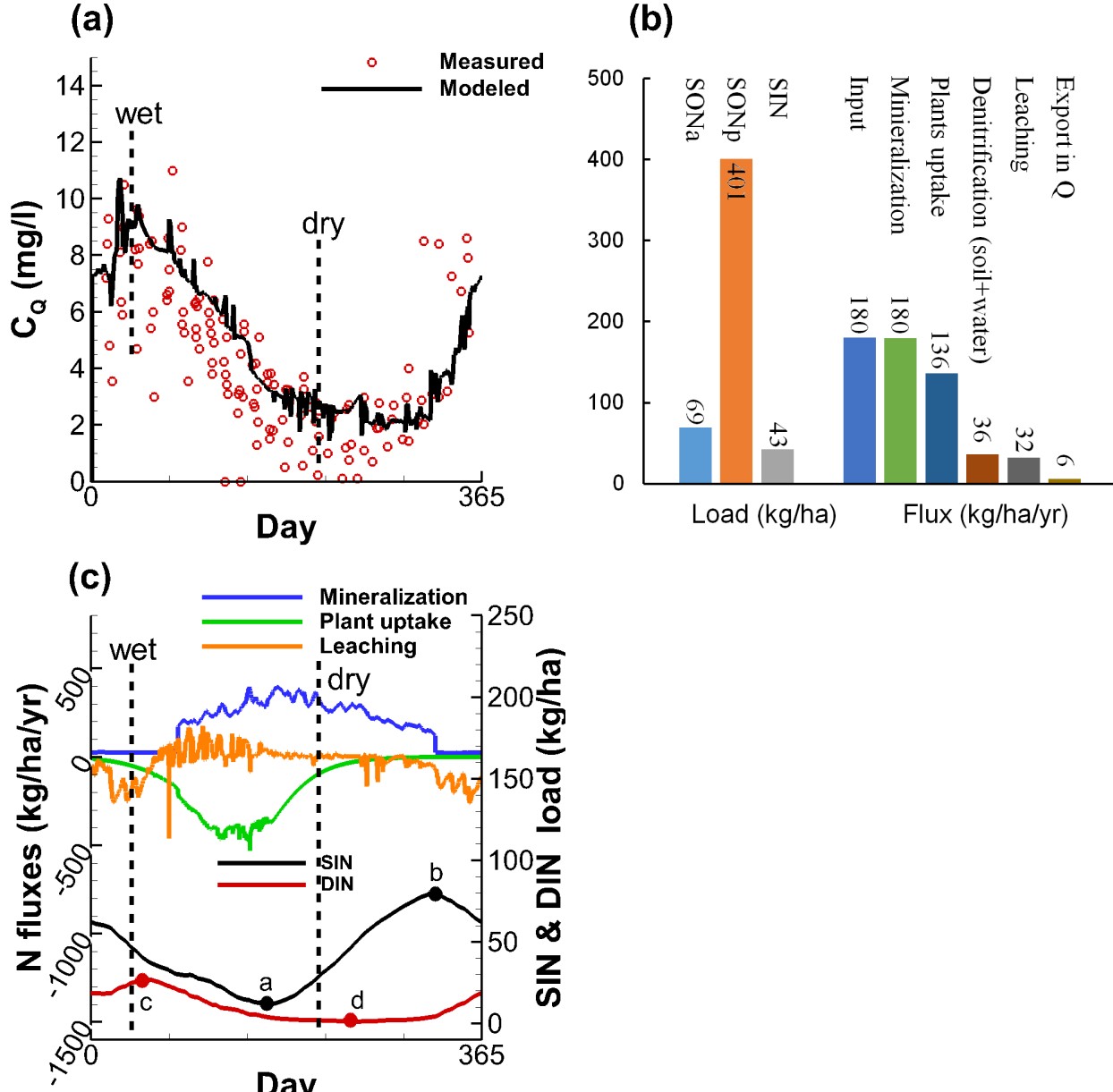

**Figure 4.** Simulated (**a**) In-stream nitrate concentration $C_Q$, (**b**) N loads and fluxes, and (**c**) time-variable N fluxes for the catchment of the base scenario. Note that the measured $C_Q$ in (a) includes all the measurements from 2001 to 2010.





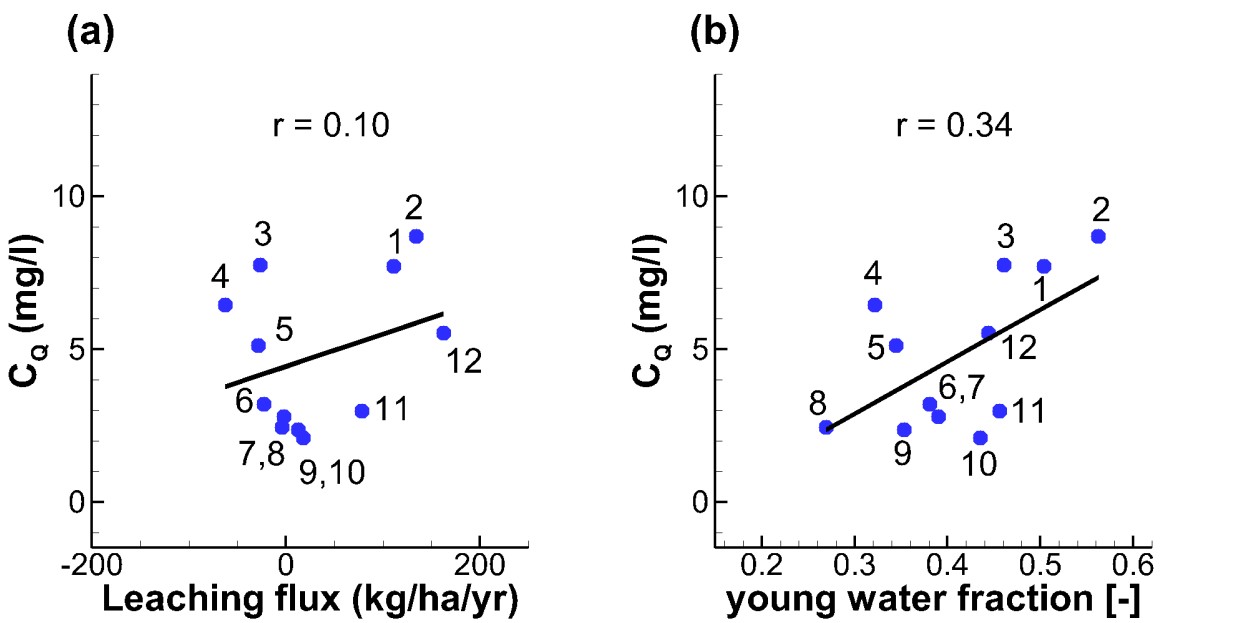

**Figure 5.** Comparing the monthly averaged $C_Q$ with (**a**) the leaching flux and (**b**) the young water fractions of Q. The black lines are linear fits of the two variables, with $r$ being the Spearman rank-correlation coefficient. The numbers refer to the months.

### 4.2 Effect of topographic slope on flow

With the help of our simulations, it is possible to systematically explore the influence of topographic slope on the water flow and N fluxes. Figure 6 shows the responses of temporally-averaged Q and ET, the groundwater table depth, and flow weighted mean $YF_Q$ and $YF_{ET}$ to the changes of topographic slope. Under a constant climate, the changes of topographic slope can reshape the water flow via influencing flow partitioning between Q and ET. More water is taken up by ET and less water becomes Q in flatter landscapes (Figure 6a). These patterns can be explained by the change of groundwater table depth (Figure 6b), as shallower groundwater tables can be reached by the vegetation in flatter landscapes where ET therefore has a higher chance to remove water from the subsurface. The simulated $YF_Q$ and $YF_{ET}$ show generally increasing and decreasing patterns, respectively, when the topographic slope decreases (Figure 6c), demonstrating that young streamflow is more prevalent in flatter landscape and young ET is more prevalent in steeper landscapes. However, the increasing pattern of $YF_Q$ does not continue in steep catchments with slopes > 1:60. Topographic slope changes $YF_Q$ not only in terms of its mean value, but also in terms of its temporal variation. Figure 6d indicates that the maximum and minimum $YF_Q$ are reached in February and August for the steepest catchment (slope 1:20), respectively, and in November and April for the flattest catchment (slope 1:1000).

458

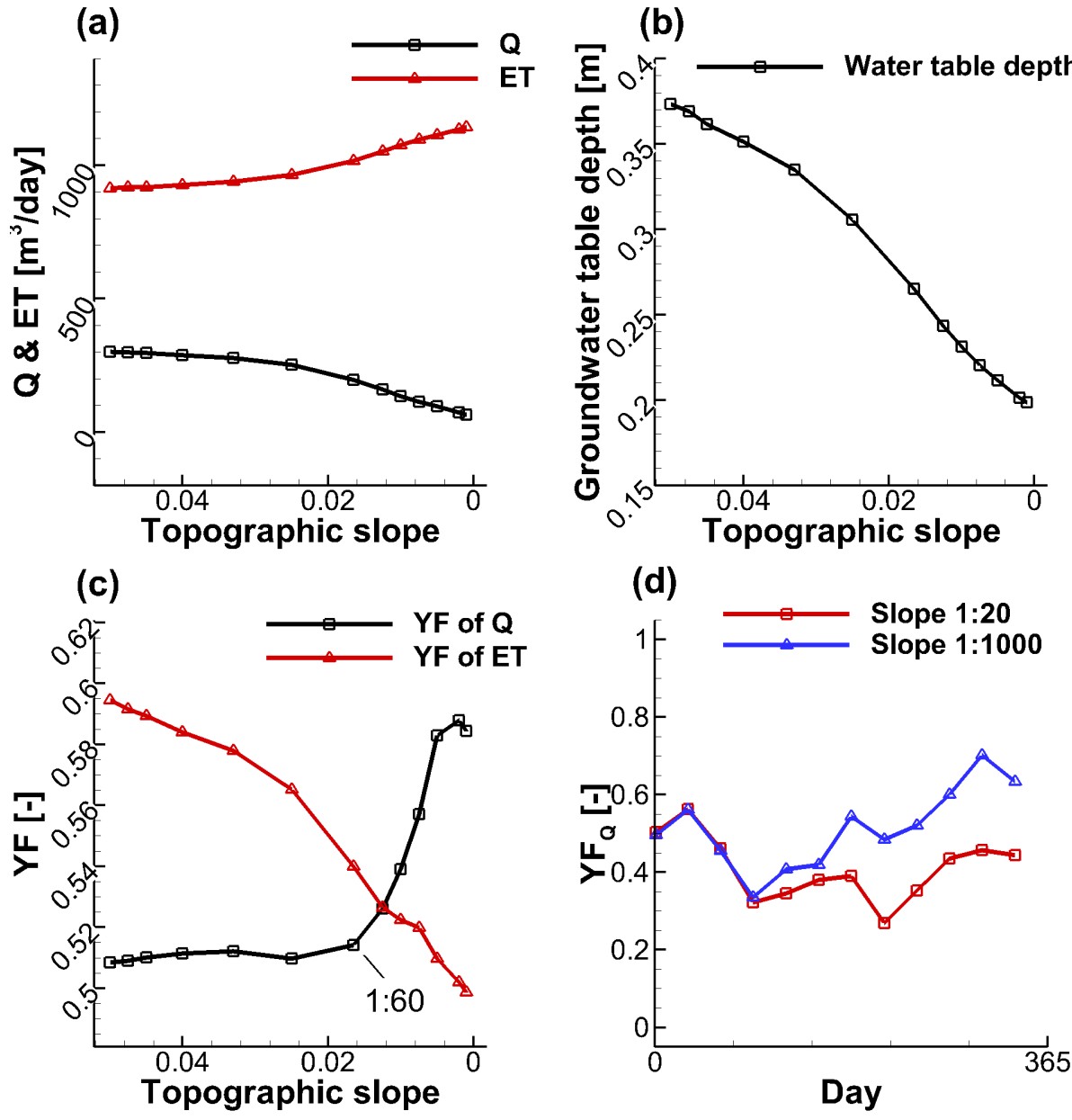

**Figure 6.** The simulated (**a**) Q and ET, (**b**) spatially-averaged depth of the groundwater table from the land surface, (**c**) young water fraction in streamflow $YF_Q$ and evapotranspiration $YF_{ET}$, in relation to the topographic slope for the simulated catchments. (**d**) temporal variations of $YF_Q$ for a steep landscape (slope 1:20) and a flat land scape (slope 1:1000).

Interpreting the response of the $YF_Q$ to topographic slope mechanistically requires a closer look at the flow processes using a cross-sectional view. We plotted the subsurface flow fields for the wet season at a cross-section of the catchments with slopes 1:20 and 1:1000 (Figure 7).

Figure 7a reveals that the hillslope part of the catchment with a slope of 1:20 is largely unsaturated so that the flow
paths in this area are characterized by vertical infiltration. In contrast, the valley bottom is fully saturated. Overall,
34% of the subsurface domain is characterized by vertical flow (flow in 34% of the total aquifer volume is more
vertical than horizontal). For this scenario two main discharge routes to the stream can be identified: (i) A fraction of
the groundwater flows through the fully saturated zone and exits the aquifer to the stream, and (ii) another fraction
exits the aquifer via seepage near to where the groundwater table intersects the land surface, indicated by a large
exchange flux (from subsurface to surface, positive). The seepage represents a preferential flow path allowing for
discharge via overland flow instead of discharge via the sub-surface with longer transit times. Note that both of the
discharge routes provide the pathways for the rainfall falling on the top hillslope to reach the stream.
When the slope is reduced to 1:1000, the flow pattern experiences significant changes (Figure 7b) compared to the
catchment with a slope of 1:20. Several hydrologic studies have described two different flow systems in aquifers: (i)
a recharge-limited system where the thickness of the unsaturated zone is sufficient to accommodate any water-table
rise and thus the elevation of the groundwater table is limited by the recharge, and (ii) a topography-limited system
where the groundwater table is close or connected to the land surface such that any fluctuation in groundwater table
can result in considerable change in surface runoff [*Werner and Simmons*, 2009; *Michael et al.,* 2013]. In the selected
cross sections, the steeper one (slope 1:20) is a partially topography-limited system (Figure 7a) (the hillslope is
recharge-limited while the valley bottom is topography-limited). The flat one (slope 1:1000) is transformed into a
fully recharge-limited system (from Figure 7b) due to the reduced hydraulic head gradients. This transformation leads
to three main effects: (i) The seepage flow vanishes because the groundwater table disconnects from the land surface.
The seepage route that would discharge water from the top of hillslope to the stream is cut off, (ii) the infiltration
processes is weakened, indicated by the fact that the portion of subsurface domain characterized by vertical flow is
reduced from 34% to 18%, and (iii) local flow cells are more likely to form, where water infiltrates to the aquifer and
eventually exits the aquifer via ET rather than via flow to the stream (Figure 7b, the local flow cells are more
pronounced in the dry season, see Figure S3-b in the supporting information).
Because of the three aforementioned effects, the connectivity between the stream and the more distant hillslopes is
significantly reduced. Precipitation falling farther from the stream has a lower chance to reach the stream and a higher
change to be intercepted by ET on its way to the stream. The hillslope that used to generate old streamflow does not
contribute to streamflow anymore. While precipitation water close to the stream has a higher chance to contribute to
streamflow. We concluded that the increase of the $YF_Q$ in flat landscapes is due to this reduction of the longer flow
paths and the persistence of shorter flow paths, as indicated by the computed TTDs (Figure 7c).

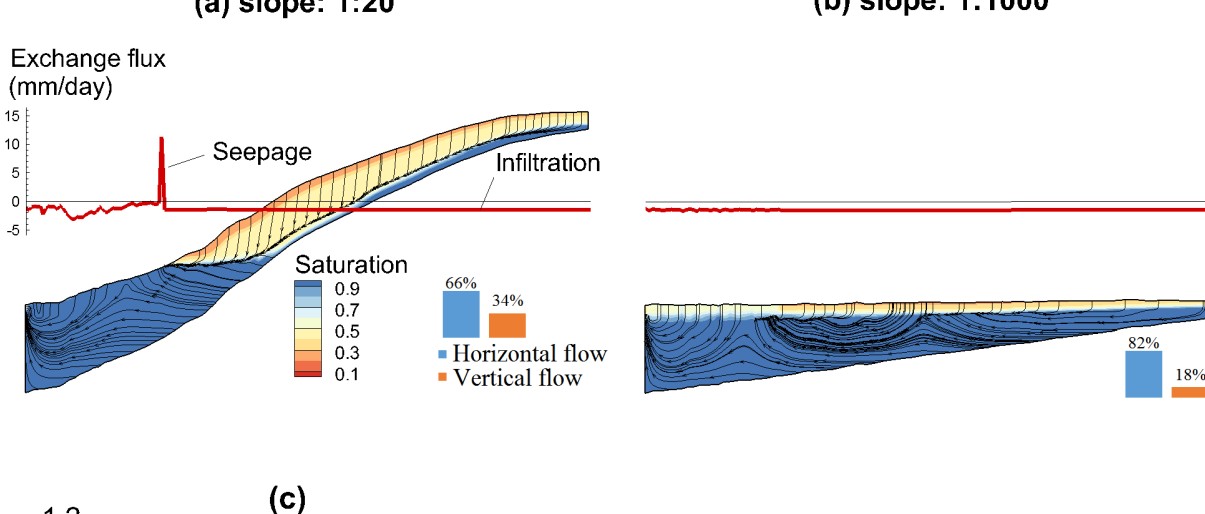

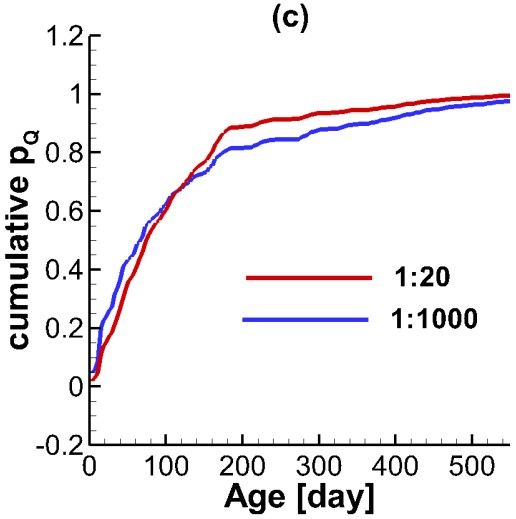

**Figure 7.** Cross-sectional view of saturation, flow paths, and exchange fluxes between the surface and the subsurface in the wet season (February) for catchments with topographic slope (**a**) 1:20, and (**b**) 1:1000. The cross-section is marked in Figure 1a. The black lines represent the flow paths. The red curves show exchange fluxes (along the cross-sectional profiles), positive values indicate seepage to the land surface and negative values indicate infiltration to the subsurface. (**c**) The computed cumulative TTDs for Q during the wet season (February), for the catchment with topographic slope of 1:20 and 1:1000.

In summary, we identified a generally increasing pattern of $YF_Q$ in response to the decreasing topographic slope. When the landscape becomes flatter, the hydraulic head gradient as the main driving force changes the aquifer from a partially topography-limited system to a recharge-limited system that is more likely to form local flow cells.

**4.3 Effect of topographic slope on N export**

Simulated results show that the topographic slope can influence the N loads and fluxes in catchments. Figure 8a demonstrates that SIN tends to be higher in flatter and lower in steeper landscapes. This generally indicates that a flat

landscape has a higher potential to retain N in the soil. However, the DIN is not significantly influenced by the topographic slope. N fluxes of leaching and export to the stream exhibit the opposite pattern. For the N fluxes, the leaching into groundwater decreases with the decrease of topographic slope (Figure 8b). This is mainly because the flow velocity (influencing the leaching rate according to equation 6) in flatter landscape is lower due to the reduced hydraulic head gradient. Comparing the time-variable leaching between the steepest and flattest catchments (slope 1:20 and 1:1000, Figure 8c), it can be observed that the leaching reduction in the flatter landscape mainly occurs in the wetting period (Nov to Dec). This may be because the response of flow velocity in the flatter catchment is not as large as that in the steeper catchment when the system transitions from dry to wet conditions. A large portion of the leached N mass has been degraded during transport in the groundwater, with the fraction rising from 80% in the steepest landscape to 95% in the flattest landscape (Figure 8b). Mechanically, the reduced connectivity between the stream and more distant hillslopes in flatter landscapes inhibits the N export to the stream promoting the degradation by increasing the N residence time in the catchment. Subsequently, the N export shows a decreasing pattern with the decrease of topographic slope (Figure 8b).

The calculated flow-weighted mean $C_Q$ shows a decreasing trend in response to the decreasing topographic slope (Figure 8d), from 7.3 mg l$^{-1}$ in the steepest catchment to 4.2 mg l$^{-1}$ in the flattest catchment. Even though both Q and N export show decreasing patterns with the decrease of topographic slope, the N export decreases to a higher degree than Q, indicated by the normalized values (Figure 8e). Comparing the time-variable $C_Q$ between the steepest and flattest catchments (slope 1:20 and 1:1000, Figure 8f), it can be observed that the topographic slope influences the $C_Q$ in two ways: (i) The $C_Q$ is generally lower (but not always) in the flatter landscape over most of the time in a year, and (ii) the high peaks of $C_Q$ in flatter landscapes are delayed in time. However, the high concentrations always occur in the wet periods (Jan – Apr) and low concentrations always occur in the dry periods (Jul – Oct).

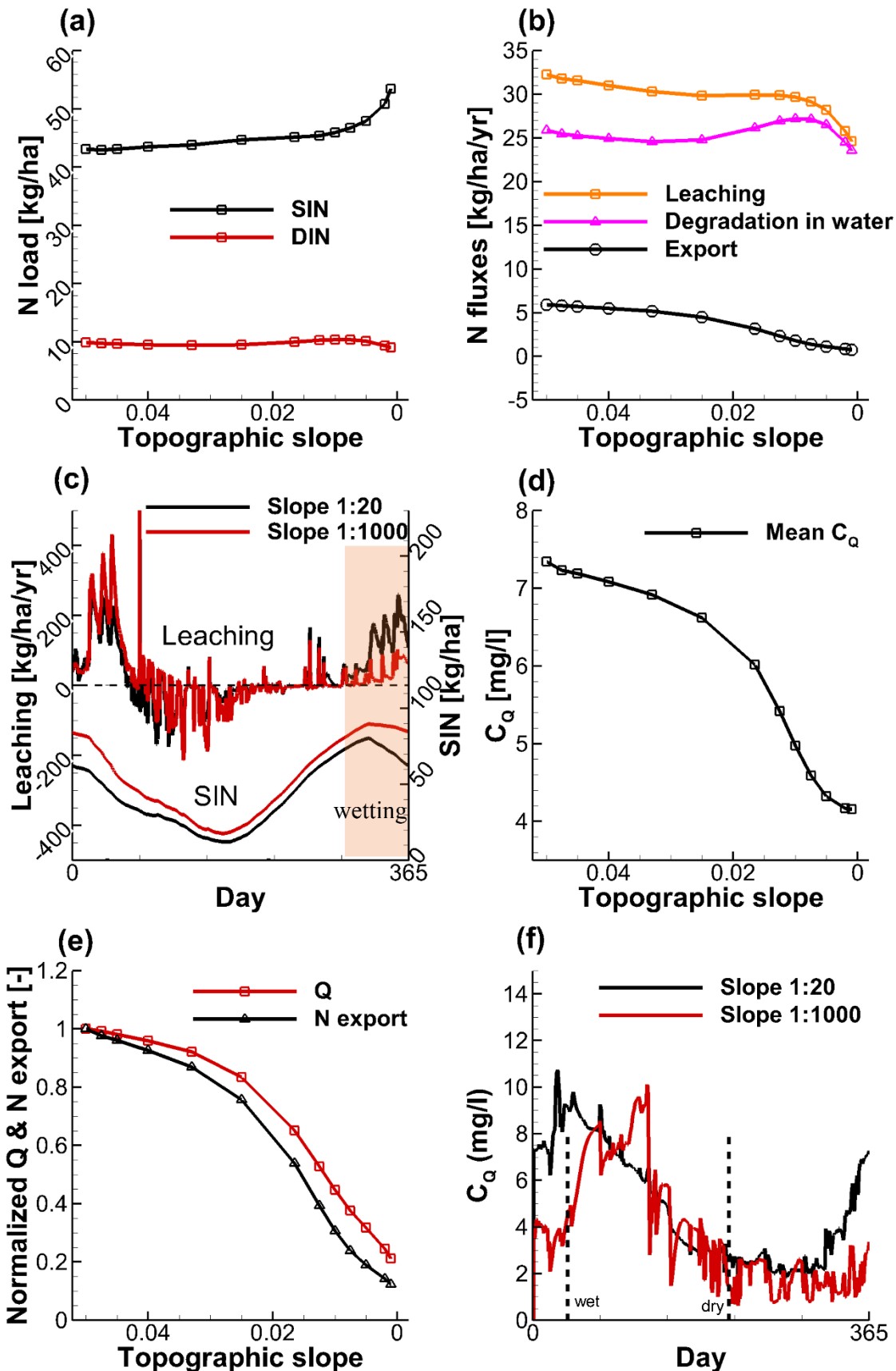

**Figure 8.** The simulated (**a**) N loads, (b) N fluxes in relation to the topographic slope for the simulated catchments.
(**c**) Comparison of the time variable N loads and fluxes between a steep (slope 1:20) and a flat land scape (slope
1:1000). The simulated (d) flow-weighted mean $C_Q$, and (e) the normalized Q and N export (normalized to their values
of the base scenario) in relation to the topographic slope. (**f**) Comparison of the time variable $C_Q$ between a steep
(slope 1:20) and a flat land scape (slope 1:1000). Note that for the leaching fluxes in (**c**), positive values are referred
to as the N leaching from the soil to the groundwater, negative values are referred to as the precipitation of N from
groundwater to the soil by the evapoconcentration effect. The vertical dashed lines indicate the time when the
catchment reaches the wettest (left) and the driest (right) conditions.

## 4.4 Discussion

*Jasechko et al*., [2016] reported that (the logarithm of) catchment topographic slope was significantly negatively
correlated with young streamflow fractions with a spearman rank correlation of -0.36. This conclusion was made
statistically based on their observed 254 sites. Our numerical study based on the eleven catchments with different
slopes but identical climate conditions resulted in more physically-based information that goes beyond such statistical
correlations. Our results confirm that young streamflow fraction and slope generally exhibit a negative correlation.
Additionally, our results show that the young water fraction in ET is positively correlated with the slope.
From the steepest landscape to the flattest landscape, catchments are likely to transition from a partially topography-
limited flow system to a recharged limited system, due to the reduction of hydraulic gradient. The groundwater table
is closer to the land surface when the landscape becomes flatter. The larger young streamflow fraction in flatter
landscapes is consistent with the statement made by *Jasechko et al*. [2016] that the young streamflow fraction is more
prevalent in flatter catchments which are characterized by more shallow lateral flow and less vertical infiltration. This
phenomenon is also consistent with a negative correlation between groundwater table depth and young streamflow
fraction, which has been frequently reported [*Bishop et al., 2004; Seibert et al., 2009; Frei et al., 2010; Jasechko et*
*al., 2016*]. Using the insight into the flow processes of the catchment, we found that the connectivity between the
stream and the more distant hillslopes is reduced in flatter landscape, due to the reduced seepage flow, the weakened
infiltration and the formation of local flow cells that do not deliver flow to the stream. Our study points out that the
reduction of this connectivity, which results in the reduction of the longer flow paths and the persistence of shorter
flow paths, causes the increase of the young streamflow fraction.
Basically, the position of the groundwater table, flow path lengths and flow velocities, which are all different for
different topographic slopes, jointly affect the young streamflow fractions. Besides that, temporal variability of these
three factors drives the distinct responses of the young streamflow fraction to topographic slope between seasons. In
our simulated catchments, the negative correlation between young streamflow fraction and topographic slope is more
pronounced in the flat landscapes with slopes < 1:60. This demonstrates that the system is complex and apparently
contains various threshold effects disturbing a straightforward monotonous relationship between catchment
characteristics (e.g. slope) and young water fraction (or streamflow concentration). In this sense, systematically
investigating the reaction of the flow dynamics to catchment characteristic is necessary, rather than assuming a
straightforward cause-effect relationship that can be misleading.
Our results demonstrate that stream water quality is potentially less vulnerable in flatter landscapes. The flatter
landscapes tend to retain more N mass in the soil and export less N mass to the stream. This behavior can be attributed
to (i) the reduced leaching in flat landscapes since the decreased flow velocity physically reduces the potential of water
to solve and transport the solute, and (ii) the increased potential of degradation because the connectivity between the
stream and hillslope is blocked (i.e. there is more time for decay). Our results also show that higher $C_Q$ is more
prevalent in steeper landscapes. Note that this is concluded for average concentrations. Observations from the Selke
catchment, central Germany show that the $C_Q$ is not always lower in flatter regions [*Dupas et al.*, 2017; Nguyen *et al*.,
2022]. In the future more attention should be paid to the temporal variation and the time-scale concerning the effect
of topographic slope on $C_Q$. Additionally, our results show that we can expect lower $C_Q$ and higher young streamflow
fractions in flatter landscapes. This suggests that, with regard to the N transport in catchments, a large young
streamflow fraction is not sufficient for high levels of $C_Q$. This phenomenon has not yet been reported to the best of
our knowledge.
Concerning the seasonal variations of $C_Q$, our results showed that significant seasonal variation can be expected under
temperate humid climates regardless of topographic slope. The high peak concentrations occurred in the wet and the
low in the dry seasons, being consistent with the findings of previous studies [*Benettin et al.* 2015; *Harman*, 2015;
*Kim et al*., 2016; *Yang et al*., 2018]. However, the topographic slope can slightly shift the high peak concentrations in
time.

**4.5 Limitations and outlook**
The cross-comparison between catchments with differing topographic slopes provides physically-based insights into
the effects of topographic slope on nitrate export responses in terms of N fluxes and mean concentrations. However,
this study is limited in scope in that it neglects other factors that may also have important impacts on the young
streamflow and nitrate export processes:
First, our study only considered the aquifers that is unconfined with an impermeable base and prescribed heterogeneity.
Other catchment characteristics such as landscape aspect, catchment area, aquifer permeability or drainage ability,
aquifer depth, stream bed elevation, fractured bedrock permeability, bedrock slope and shape of basin can potentially
change the flow patterns and age composition in streamflow [*McGlynn et al.,* 2003*; Broxton et al.,* 2009; *Sayama and*
*McDonnell*, 2009; *Stewart et al*., 2010; *Jasechko et al*., 2016; *Heidbüchel et al*., 2013, 2020; *Zarlenga and Fiori*,
2020]. For example, aquifers with high permeability or highly fractured bed rock are more likely to use deep rather
than shallow flow paths and preferential discharge routes that lead to rapid drainage. Apart from that, it was reported
that hydrological features such as precipitation variability, ET, antecedent soil moisture are also significantly linked
to transit times [*Sprenger* et al., 2016; *Wilusz et al.* 2017; *Evaristo et al*., 2019; *Heidbüchel et al*., 2013, 2020]. For
example, compared to uniform precipitation, event-scale precipitation is more likely to trigger rapid surface runoff
and intermediate flow, such that the contribution of young water from storage to streamflow can be increased.
Therefore, further research should consider a more complex model structure involving various heterogeneity and
climate types.
Second, several main simplifications were used in the formulation of the nitrate transport processes. (i) Transport
modelling employed a constant degradation rate coefficient assuming that transit time was the only factor to determine
degradation. This assumption neglected other factors that can spatially and temporally affect denitrification rates, such
as temperature, redox boundaries (e.g., high oxygen concentration in shallow flow paths), the amount of other nutrients
(e.g. carbon), which also contribute to the seasonality in nitrate concentrations [*Böhlke et al.*, 2007]. Apart from that,
we did not account for the long-term (decades [*Van Meter et al.*, 2017]) nitrate legacy effect as the dissolved nitrate
in groundwater reservoirs degraded continuously in our model, which would not occur in older reservoirs where the
denitrification is very slow or deactivated (e.g. due to the lack of a carbon source). (ii) The N external input source
was uniformly applied across the land surface in our modelling. However, strong source heterogeneity may exist in
catchments. For example, the N external input varies between land uses or along the soil profile [*Zhi et al.*, 2019].
This spatial source heterogeneity could affect the seasonal variations of $C_Q$ [*Musolff et al.*, 2017; *Zhi et al.*, 2019] and
should be considered in further research.
While the numerical model provided general insights, there was potential uncertainty in the simulated results. Firstly,
the aforementioned simplifications may introduce model structural errors. Secondly, the model calibration was only
constrained by limited data sets, which may lead to the non-uniqueness in the model parameters. Both of the aspects
may introduce uncertainty in the simulated N loads and fluxes. Future work should be devoted to better constrain the
model parameters, either by enhancing the concentration data quality through more frequent measurements or by
providing additional data sets related to the N pool. Despite these limitations, the numerical experiments in this study
could clearly identify the response of young streamflow and nitrate export to topographic slope under a humid seasonal
climate, and show that hydraulic gradient is an important factor causing flow field differences between the catchments.
This was achieved by using the advantages of a physically-based flow simulation that allows for a more mechanistic
evaluation of flow processes, which would be impossible with a purely data driven analysis based on, e.g., isotopic
tracers only.

**5 Conclusions**
Previous data driven studies suggested that catchment topographic slope impacts the age composition of streamflow
and consequently the in-stream concentrations of certain solutes [*Jasechko et al.*, 2016]. We attempted to find more
mechanistic explanations for these effects. We chose the small agricultural catchment 'Schäfertal' in Central Germany
and, based on it, generated eleven synthetic catchments of varying topographic slope. The groundwater and overland
flow, and the N transport in these catchments were simulated using a coupled surface-subsurface model. Water age
compositions for Q and ET were determined using numerical tracer experiments. Based on the calculated flow patterns,
young water fractions in streamflow $YF_Q$, N mass fluxes and in-stream nitrate concentration $C_Q$, we systematically
assessed the effects of varying catchment topographic slopes on the nitrate export dynamics in terms of the mass fluxes
and annual mean concentration levels. The main conclusions of this study are:
● Under the considered humid climate, $YF_Q$ is generally negatively correlated to topographic slope. When the
landscape becomes flatter, the hydraulic head gradient is the main driving force to change the aquifer from a
partially topography-limited system to a recharge-limited system, reducing the connectivity between the
stream and the more distant hillslopes. This change results in the reduction of longer flow paths and the
persistence of shorter flow paths, subsequently causing the flatter landscapes to generate younger streamflow.
● The flatter landscapes tend to retain more N mass in soil and export less N mass to the stream. These patterns
are attributed to (i) the reduced leaching in flat landscape as the decreased flow velocity physically reduces
the potential of water to transport the solute towards the stream, and (ii) the increased potential of degradation
as the connectivity between the stream and hillslope is blocked and the solute stays inside the aquifer longer.
● For the considered catchment, the annual mean $C_Q$ shows a decreasing trend in response to the decreasing
topographic slope, because the N export decreases to a higher degree than Q. Flatter landscapes tend to
generate larger young streamflow fractions (but lower $C_Q$), suggesting that a large young streamflow fraction
is not sufficient for a high level of $C_Q$.
Overall, this study provided a mechanistic perspective on how catchment topographic slope affects young streamflow
fraction and nitrate export patterns. The use of a fully-coupled flow and transport model extended the approach to
investigate the effects of catchment characteristics beyond the frequently used tracer data-driven analysis. It can be
used for similar studies of other catchment characteristics and for other solutes. The results of this study improved the
understanding of the effects of certain catchment characteristics on nitrate export dynamics with potential implications
for the management of stream water quality and agricultural activity, in particular for catchments in temperate humid
climate with pronounced seasonality. Given the limitations of this study, future work should be devoted to improve
the degradation formulation, to investigate further catchment characteristics, as well as to consider various climate
types.


**Notation**
$t$    [T] time
$T$    [T] age / transit time / residence time
$J$    $[LT^{-1}]$ precipitation
$ET$    $[LT^{-1}]$ evapotranspiration
$Q$    $[LT^{-1}]$ discharge / streamflow
$p_S$    [-] age distribution of storage
$p_{ET/Q}$    [-] age distribution for evapotranspiration / discharge, equivalent to TTD
$C$    $[ML^{-3}]$ concentration
$C_Q$    [ML$^{-3}$] in-stream solute (nitrate) concentration
$T_Q$    [ML$^{-3}$] age (transit time) of discharge
$YF_Q$    [-] young water fraction in streamflow, or young streamflow fraction
$YF_{ET}$    [-] young water fraction in ET
*SON*    [M L$^{-2}$] soil organic nitrogen
*SIN*    [M L$^{-2}$] soil inorganic nitrogen
*DIN*    [M L$^{-2}$] dissolved inorganic nitrogen in water



**Code/Data availability**

All data used in this study are listed in the supporting information and uploaded separately to HydroShare [Yang,
2022].

**Author contributions**

JY: conceptualization, methodology, software, formal analysis, visualization, writing - review & editing; QW:
modelling, analysis, writing; IH: writing - review & editing; CL: conceptualization, methodology, review & editing;
YX: methodology; AM: conceptualization; JF: conceptualization, review & editing.

**Competing interests**

The authors declare that they have no conflict of interest.

**Acknowledgments**

This research was supported by the National Key Research and Development Project (JY & CL: 2021YFC3200500),
the National Natural Science Foundation of China (JY: 52009032, CL: 51879088), the Fundamental Research Funds
for the Central Universities (JY: B210202019), and the Natural Science Foundation of Jiangsu Province (CL:
BK20190023). We thank the editorial board for handling our manuscript, especially Prof. Dr. Insa Neuweiler and two
anonymous reviewers, whose constructive comments helped improve the manuscript.

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
