# Peer review of "Effect of topographic slope on the export of nitrate in humid catchments: a 3D model study"

_Hydrology and Earth System Sciences, 2021_

## Referee Comment (RC1)

This manuscript presents a series of hillslope-scale numerical experiments intended to explore how the topographic slope controls nitrate export. The authors applied a hillslope model with parameters from their previous catchment-scale modeling work and meteorology data from a small agriculture catchment in Central Germany as input. They mainly found that the response of in-stream nitrate concentrations to topographic slope is a three-class pattern rather than monotonous. The science questions and approach would appeal to the HESS journal audience and make a nice contribution to understanding these connections between topography, subsurface flow paths, and nitrate export.

However, I had some major concerns that need to be addressed prior to publication:

1) It is lack of field data to either constraint or validate the model. Although the authors provided lots of descriptions about their study site, it seems to me that they only applied the meteorology data and some parameters from the previous modelling work at this site as the modelling input. Neither the simulated hydrological nor biogeochemical (nitrate) part are justified or calibrated according to the real field data (e.g., water table depth, water content, and nitrate concentrations). Even though the authors stated that it is not important to accurately reproduce the flow discharge and solute concentration, I think it is still necessary to make sure that the simulated values are comparable to observations.

2) It was not always clear throughout the manuscript what assumptions are made, why the authors made such assumptions, and their implications. For instance, the authors used annual-average precipitation and monthly-averaged potential ET as the input to simulation the hydrological dynamics across the year. This leads to the inconsistent of time scales for precipitation and pET, especially for this work with a focus on seasonal variations. I don't understand why not to use monthly-averaged precipitation to keep consistent with pET. For the nitrate transport, to simply the model, the model does not include the evapoconcentration effect for nitrate transport. If that is the case, how does the model handle the nitrate concentration and fluxes from the input source with precipitation (i.e., before ET) to soil water after ET occurs? It seems to be mass imbalance for nitrate transport.

3) Additional results are needed to better support the conclusion. The objective of the manuscript is to explore the influence of topographic slope on nitrate export. However, the authors only showed the effluent nitrate concentration and its temporal variations. How about the overall export rate of nitrate (concentration * effluent flow rate)? I think this may better reflect nitrate export. The main reason is that the topographic slope alters the water content and water table depth (Figure 4d), and further change the simulated ET and how much water infiltrates into the subsurface and eventually enter the stream. Besides, the authors assumed (Line 220) that ET does not alter the nitration concentration in the subsurface. However, if ET varies with topographic slope, this would lead to the inconsistency of the overall nitrate mass into the subsurface for different scenarios. Therefore, the conclusions based on in-stream concentrations might not really hold true.

4) The assessment of source contribution and the terms of source- and degradation-dominated needs clarification. I remained unclear about the physical meaning of the equation the authors applied to calculate the source contribution. Line 284: why does 0 and 100% represents degradation-dominated and source-dominated, respectively? I think this is conflicted with the main assumption that nitrate transport is dominated by the transit time or flow path (i.e., hydrology-dominated). Besides, Line 284-297: the authors introduced Damköhler number in the

method but did not really use it in the rest part of the manuscript. I guess they intended to define transport- and reaction-limited system?

5) In-depth discussion about nitrate export is needed. The main goal of this work is to build a connection between hillslope topography and in-stream nitrate concentration through flow paths and water age. However, the current discussion mainly focused on the influence of hillslope topography on subsurface flow paths and water age. Their linkages with nitrate export and how the numerical results here are related to previous field and modelling work are largely missing. Besides, this work is based on hillslope-scale numerical experiments rather than catchment-scale modelling. The authors should also consider the scale differences when they directly applied results from this work to explain observations at the catchment scale.

Other comments with line number:

Line 215-219 and Figure 2: Is the constant source calculated from flow averaged or time-variant concentration averaged? I think it needs to be flow averaged. Otherwise, the overall input mass is not consistent between the two scenarios.

Figure 5: Horizontal flow and vertical flow: how to calculate the percentage? Even in the saturated zone, the water can still flow horizontally and vertically.

Figure 5 and 7: what about the spatial distributions of nitrate concentrations? I think adding such figures may help the readers understand the modelling and results.

Line 355-360 and Figure 5: The flow fluxes into the land surface take a high proportion in the case of slope 1:20 and 1:60. Does nitrate continue to degrade actively in the land surface? If it does, this may not really hold true for real nature systems. Please clarify and discuss the potential implication.

---

## Author Comment (AC1)

Dear reviewer,

please check our responses to your comments (marked in blue). The final changes will be made to the manuscript at the stage of submitting revision later. Thank you!

**RC1:**

This manuscript presents a series of hillslope-scale numerical experiments intended to explore how the topographic slope controls nitrate export. The authors applied a hillslope model with parameters from their previous catchment-scale modeling work and meteorology data from a small agriculture catchment in Central Germany as input. They mainly found that the response of in-stream nitrate concentrations to topographic slope is a three-class pattern rather than monotonous. The science questions and approach would appeal to the HESS journal audience and make a nice contribution to understanding these connections between topography, subsurface flow paths, and nitrate export.

However, I had some major concerns that need to be addressed prior to publication:

1) It is lack of field data to either constraint or validate the model. Although the authors provided lots of descriptions about their study site, it seems to me that they only applied the meteorology data and some parameters from the previous modelling work at this site as the modelling input. **Neither the simulated hydrological nor biogeochemical (nitrate) part are justified or calibrated according to the real field data** (e.g., water table depth, water content, and nitrate concentrations). Even though the authors stated that it is not important to accurately reproduce the flow discharge and solute concentration, I think it is still necessary to make sure that the simulated values are **comparable** to observations.

Response #1:

We agree that the comparison with data was missing in the study. Our 2D model was extracted from a 3D one of our precious modelling work, where the stream discharge, groundwater table were calibrated against measurements. Unfortunately, we do not have discharge and groundwater table data specially for the 2D slice.

We have nitrate concentration (C) data measured at the outlet of the 3D catchment. These C data are not for the discharge from 2D slice, however, we can still show that the simulated C is comparable with the measured C, but to exactly reproduce the measured C should not be expect. The comparison will be added to the manuscript in our revised version.

2) It was not always clear throughout the manuscript what assumptions are made, why the authors made such assumptions, and their implications. For instance, the authors used annual- average precipitation and monthly-averaged potential ET as the input to

simulation the hydrological dynamics across the year. This leads to the inconsistent of time scales for precipitation and pET, especially for this work with a focus on seasonal variations. I don't understand why not to use **monthly-averaged precipitation** to keep consistent with pET. For the nitrate transport, to simply the model, the model does not include the evapoconcentration effect for nitrate transport. If that is the case, how does the model handle the nitrate concentration and fluxes from the input source with precipitation (i.e., before ET) to soil water after ET occurs? It seems to be mass imbalance for nitrate transport.

Response #2:

Thank for the suggestion. We will change to monthly-averaged precipitation to ensure consistence in the revised version. We used constant precipitation but seasonally changing ET because the seasonality of the catchment is main driven by the ET and the precipitation is more uniformly distributed over the year.

For the N mass balance issue, we used an approach of N source concentration curve, which force the N concentration in the rainfall to change along the curve at the moment of entering the soil. The assumption was that the variation of N source concentration caused by the evapoconcentration was already considered in the N source concentration curve. This assumption did cause mass imbalance of the system. To overcome this limitation, we will use a different approach, which has been used in the study of *Yang et al.,* 2021, to simulate the nitrogen fluxes in the soil, including the input, mineralization, degradation in soil, crop-uptake and leaching into groundwater (see the figure below). This approach will increase model complexity by introducing new parameters, however, ensure a mass balance. This work will be updated to the manuscript in our revised version.

[Figure]

**(a) Source zone**

Reference:

Yang, J., Heidbüchel, I., Musolff, A., Xie, Y., Lu, C.*, Fleckenstein, J.H. (2021). *Using nitrate as a tracer to constrain age selection preferences in catchments with strong seasonality*, Journal of Hydrology, 603, 126889.

3) Additional results are needed to better support the conclusion. The objective of the manuscript is to explore the influence of topographic slope on nitrate export. However, the authors only showed the effluent nitrate concentration and its temporal variations. How about the **overall export rate of nitrate** (concentration * effluent flow rate)? I think this may better reflect nitrate export. The main reason is that the topographic slope alters the water content and water table depth (Figure 4d), and further change the simulated ET and how much water infiltrates into the subsurface and eventually enter the stream. Besides, the authors assumed (Line 220) that ET does not alter the nitration concentration in the subsurface. However, if ET varies with topographic slope, this would lead to the inconsistency of the **overall nitrate mass** into the subsurface for different scenarios. Therefore, the conclusions based on in-stream concentrations might not really hold true.

Response #3:

Thanks for the suggestion. We will discuss also the exported nitrate mass flux in the revised version to better support the conclusions.

For the mass balance issue caused by the ET. We will use a different approach to simulate the nitrogen fluxes in the soil (see the last response). This approach will ensure that the input of overall nitrogen mass into the subsurface is same for all scenarios.

4) The assessment of source contribution and the terms of source- and degradation-dominated needs clarification. I remained unclear about the physical meaning of the equation the authors applied to calculate the source contribution. Line 284: why does 0 and 100% represents degradation-dominated and source-dominated, respectively? I think this is conflicted with the main assumption that nitrate transport is dominated by the transit time or flow path (i.e., hydrology-dominated). Besides, Line 284-297: the authors introduced Damköhler number in the method but did not really use it in the rest part of the manuscript. I guess they intended to define transport- and reaction-limited system?

Response #4:

Thanks for pointing out that. What we want to express was that the variation of in-stream concentration may attribute to two aspects: (i) the availably of nitrite source changes over time, and (ii) the reaction (degradation) along the flow path. The term "Degradation-dominated" may be more accurate expression than the term "hydrology-dominated", because the degradation is a combination of transit time (influenced by flow paths) and degradation rate (influenced by various factors such as temperature). We will clarify that in the revised version.

Thanks, we will simply delete the unused content of Damköhler in the revised version.

5) In-depth discussion about nitrate export is needed. The main goal of this work is to build a connection between hillslope topography and in-stream nitrate concentration through flow paths and water age. However, the current discussion mainly focused on the influence of hillslope topography on subsurface flow paths and water age. Their linkages with nitrate export and how the numerical results here are related to previous field and modelling work are largely missing. Besides, this work is based on **hillslope-scale numerical experiments r**ather than catchment- scale modelling. The authors should also consider the s**cale differences** when they directly applied results from this work to explain observations at the catchment scale.

Response #5:

We agree with that. We will add more discussion about the nitrate fluxes between different scenarios. We will try to relate our simulated result with previous studies. We will also clarify that our study was based on the hillslope-scale and discussion the influences of the scale differences. This work will be updated to the manuscript in our revised version.

Other comments with line number:

Line 215-219 and Figure 2: Is the constant source calculated from flow averaged or time-variant concentration averaged? I think it needs to be flow averaged. Otherwise, the overall input mass is not consistent between the two scenarios.

Response #6:

We used a time-weighted average rather flow-weighted average. Because we used a constant rainfall over the year, that means same amount of nitrate was inputted into the system. However, we plan to use a different approach to define the nitrate source zone (see response #2), replacing the nitrate source concentration curves. This work will be updated to the manuscript in our revised version.

Figure 5: Horizontal flow and vertical flow: how to calculate the percentage? Even in the saturated zone, the water can still flow horizontally and vertically.

Response #7:

For each cell, we check the velocity to see if it is more horizontal or more vertical. Then we summed up all the volumes of all cells where velocity is more horizontal (or vertical) and calculate the portion to total aquifer volume. We will clarify that in the revised version.

Figure 5 and 7: what about the spatial distributions of nitrate concentrations? I think adding such figures may help the readers understand the modelling and results.

Response #8:

Thanks for the suggestion, we will add that figure during the revision.

Line 355-360 and Figure 5: The flow fluxes into the land surface take a high proportion in the case of slope 1:20 and 1:60. Does nitrate continue to degrade actively in the land surface? If it does, this may not really hold true for real nature systems. Please clarify and discuss the potential implication.

Response #9:

Thanks for pointing out that. The degradation was not considered on land surface. We will clarify that in the revision.

---

## Author Comment (AC2)

Dear reviewer,
please check our responses to your comments (marked in blue). The final changes will be made to the manuscript at the stage of submitting revision later. Thank you!

**RC2:**

Overall, this is a nice contribution seeking to explore the impact of the slope of a hillslope transect on nitrate transport and export with the aid of numerical simulations. The analysis is developed using a coupled numerical model for water and solute transport, which also simulates water ages. The topic is of interest to the readership of HESS and I think the Ms could make a good addition to the literature. The text is in general well written and properly organized. However, there are a few limitations that I would like to emphasize in what follows.

I think this is not the first study to explore the impact of the slope of water ages using numerical tools (e.g. Zarlenga and Fiori, 2020), nor the first study to model the export of nitrogen in the context of water ages (van der Velde et al., 2012; Benettin et al., 2020) so I would better put this work in the context of the state of the art. Jasechko et al., 2016 should not be the only conerstone for this study, as it seems to be at times.

Response #1:

Thanks for pointing out that! We will correct the statements in the introduction and add the previous studies in the state of the art. This update will be made in the revised version of this manuscript.

**The lack of empirical data** to constrain the underlying model parameters is a little bit worrisome. I understand it is difficult to have a full comprehensive analysis of the uncertainty owing to significant computational times, but the authors should put more effort in demonstrating that their simulations are a **reasonable representation of the real world**. I would add more simulations under different scenarios in terms of model parameters, trying to identify how the results obtained in the paper could change if some settings of the numerical simulations are modified (e.g. profile likelihood, sensitivity analysis). A lot of parameters are simply assumed a priori.

Response #2:

Thanks for the suggestions. We will try to link our simulated nitrate concentration with the measurements, to show that they are comparable. We will add sensitivity analysis for the parameters (e.g., profile length). These parts will be updated during the revision.

The way evapotranspiration is treated in the transport model is not described in detail. This is a key process in this context (e.g. changes in the uptake depth of the roots might have a strong impact on the results in some cases) and more emphasis should be given

to describe how the numerical code models the green water.

Response #3:

Thanks for the suggestions! We will clarify the computation of ET in the HydroGeoSphere model and add contents to describe the simulation of water flow. These updates will be made during the revision.

Nitrate is here described as a decaying solute, but I'm not fully convinced by the explanation given by the authors to justify their approach. In particular, I'm not sure that **biogeochemical processes** other than degradation that take place during the transport processes along the hillslope can be completely ignored (i.e. treated as an off-line mechanism that impacy only the initial condition C_J) especially if the solute export is the final goal of the study. More emphasis should be given to the export in the paper as compared to the "transport" issue.

Response #4:

Thanks for pointing out that! We used an approach of N source concentration curve, which force the N concentration in the rainfall to change along the curve at the moment of entering the soil. Using this approach the biogeochemical processes happened in the nitrogen source zone was ignored. This approach did cause mass imbalance of the system. To overcome this limitation, we will use a different approach, which has been used in the study of *Yang et al.,* 2021, to simulate the nitrogen fluxes in the soil, including the input, mineralization, degradation in soil, crop-uptake and leaching into groundwater (see the figure below). This approach will increase model complexity by introducing new parameters, however, ensure a mass balance. This work will be updated to the manuscript in our revised version.

[Figure]

Reference:

Yang, J., Heidbüchel, I., Musolff, A., Xie, Y., Lu, C.*, Fleckenstein, J.H. (2021). *Using nitrate as a tracer to constrain age selection preferences in catchments with strong seasonality*, Journal of Hydrology, 603, 126889.

Generalizability issues should be disccused more deeply. How these results might apply to other settings beyond the specific case study presented in the MS and the role of the 3D complexity of a catchment, which is not modeled here? Why do the authors believe their findings are general?

Response #5:

We agree with that! We will be very careful when descript our conclusion found in the specific case in a general way. We will also clarify that our study was based on the hillslope-scale and discussion the influences of the scale differences. This work will be updated to the manuscript in our revised version.

Minor points

96: a fairer chain of references here – especially if you talk about TTD of ET - should be Botter et al., 2010; 2011; Van der Velde et al., 2012; Rinaldo et al., 2015, Harman et al., 2015, 2019.
123-124: this seems to be somewhat speculative at this stage. Move to the discussion

and elaborate plz.

150: maybe "Climate" instead of "Climates"?

165: plz explain in the caption the motivation for the 6 shaded regions represented in panel b of the Figure

195-205: plz provide more details about the boundary conditions at the bottom of the domain.

260: not sure this is correct. Plz double check. Why a Deltat is needed from the physical viewpoint? This should be something in the continuous-time domain. Morevoer, the T should appear also on the r.h.s. of the equation (I see there is some text on this in the following lines but I would polish the expression a little bit to make it consistent with the existing literature).

Figure 4: mean should be intermediate here

Line 430: what about continuous instead of monotonous?

Response #6:

Thanks for the minor comments. They will be carefully responded and corrected correspondingly in our revised version at the stage of submitting the revised manuscript.

---

## Author Response (AR1)

Dear reviewers and editor,
please check our responses to your constructive comments (marked in blue). Unless specified, all the line numbers in our responses refer to the line numbers of the TRACK CHANGE version of the manuscript. Thank you!

Response overview:

According to the reviewers' and the editor's comments, we have made updates to the manuscript, main key updates are:
1. We changed the modelling site from a 2D hillslope to a 3D catchment. We added 'a 3D model study' to the title. The measured data for the 3D catchment is used for comparing with the modeled ones, including nitrate concentration measured at the outlet and the nitrogen surplus data.
2. We employed a different framework, which has been used in the study of *Yang et al., 2021*, to simulate the nitrogen fluxes in the soil, including the input, mineralization, degradation in soil, crop-uptake and leaching into groundwater. This framework had increased model complexity by introducing new parameters, however, it ensured a correct mass balance and delivered nitrogen fluxes in the soil.
3. The results part was updated. In addition to the discussion about the young water fractions and nitrate concentrations in stream, we included further discussion about the effect of topographic slope on nitrogen fluxes, including leaching, degradation, and export to the stream. We tried to link our simulated results with other previous studies to show that our model is a reasonable representation of the real world catchment under the temperate humid climate. In this way, the conclusion made in this study is more likely to apply to other catchment under similar climate.

**RC1:**

This manuscript presents a series of hillslope-scale numerical experiments intended to explore how the topographic slope controls nitrate export. The authors applied a hillslope model with parameters from their previous catchment-scale modeling work and meteorology data from a small agriculture catchment in Central Germany as input. They mainly found that the response of in-stream nitrate concentrations to topographic slope is a three-class pattern rather than monotonous. The science questions and approach would appeal to the HESS journal audience and make a nice contribution to understanding these connections between topography, subsurface flow paths, and nitrate export.

However, I had some major concerns that need to be addressed prior to publication:

1) It is lack of field data to either constraint or validate the model. Although the authors provided lots of descriptions about their study site, it seems to me that they only applied the meteorology data and some parameters from the previous modelling work at this site as the modelling input. **Neither the simulated hydrological nor biogeochemical (nitrate) part are justified or calibrated according to the real field data** (e.g., water table depth, water content, and nitrate concentrations). Even

though the authors stated that it is not important to accurately reproduce the flow discharge and solute concentration, I think it is still necessary to make sure that the simulated values are **comparable** to observations.

Response #1:

We agree that the comparison with data was missing in the study. That was because we did not have any data specially for the 2D hillslope of the catchment. In the revision, we changed the modelling site from the 2D hillslope to a 3D catchment. The measured data for the 3D catchment is used to compare with the modeled ones, including nitrate concentration measured at the outlet (Figure 4 and Figure S1 in supporting information) and the nitrogen surplus data (line 367-369). The simulated N fluxes and loads in soil were verified by comparing with literature values (line 457-474). These demonstrated that that our simulated values were fit well or comparable to the observations and our model is a reasonable representation of the real world.

2) It was not always clear throughout the manuscript what assumptions are made, why the authors made such assumptions, and their implications. For instance, the authors used annual- average precipitation and monthly-averaged potential ET as the input to simulation the hydrological dynamics across the year. This leads to the inconsistent of time scales for precipitation and pET, especially for this work with a focus on seasonal variations. I don't understand why not to use **monthly-averaged precipitation** to keep consistent with pET. For the nitrate transport, to simply the model, the model does not include the evapoconcentration effect for nitrate transport. If that is the case, how does the model handle the nitrate concentration and fluxes from the input source with precipitation (i.e., before ET) to soil water after ET occurs? It seems to be mass imbalance for nitrate transport.

Response #2:

Thanks for the suggestion. We changed both the precipitation and pET to daily resolution in the revision.

For the N mass balance issue, we used a different framework, which has been used in the study of *Yang et al.,* 2021, to simulate the fate of N in the N pool (shallow soil zone), including the external N input, mineralization, degradation in soil, crop-uptake and leaching into groundwater (Line 266-308, see following figure). The evapoconcentration effect was considered in this framework by allowing ET to remove DIN mass from water and to inject that mass back to the SIN pool in soil (line 326-332). This framework increases the model complexity by introducing new parameters, however, it also ensures a correct mass balance.

[Figure]

**(a) Soil (N pool)**

Input (atmospheric, biological fixation, manure, fertilizer)

SON (active)  SON (protected)

Mineralization

SIN

Plants uptake    Denitrification (soil)

Leaching

**(b) Groundwater**

Precipitation

DIN

Denitrification (water)

Export to stream

(from Figure 2 of the revised manuscript)

Reference:

Yang, J., Heidbüchel, I., Musolff, A., Xie, Y., Lu, C.*, Fleckenstein, J.H. (2021). *Using nitrate as a tracer to constrain age selection preferences in catchments with strong seasonality*, Journal of Hydrology, 603, 126889.

3) Additional results are needed to better support the conclusion. The objective of the manuscript is to explore the influence of topographic slope on nitrate export. However, the authors only showed the effluent nitrate concentration and its temporal variations. How about the **overall export rate of nitrate** (concentration * effluent flow rate)? I think this may better reflect nitrate export. The main reason is that the topographic slope alters the water content and water table depth (Figure 4d), and further change the simulated ET and how much water infiltrates into the subsurface and eventually enter the stream. Besides, the authors assumed (Line 220) that ET does not alter the nitration concentration in the subsurface. However, if ET varies with topographic slope, this would lead to the inconsistency of the **overall nitrate mass** into the subsurface for different scenarios. Therefore, the conclusions based on in-stream concentrations might not really hold true.

Response #3:

Thanks for the suggestion.
We discussed the effect of topographic slope on N export also in terms of the mass flux with Figure 8b, showing that the N export shows a decreasing pattern with the decrease of topographic slope (Line 684-698).

For dealing with the mass balance issue caused by the ET, we will use a new framework to simulate the nitrogen fluxes in the soil (see the response #2). This framework ensures that the input of overall nitrogen mass into the subsurface is the same for all scenarios. With this framework, ET does not cause mass imbalance because the mass taken up by ET is injected back to the SIN pool, representing the evapoconcentration effect that causes the precipitation of nitrate from water (DIN) to the soil (SIN).

4) The assessment of source contribution and the terms of source- and degradation-dominated needs clarification. I remained unclear about the physical meaning of the equation the authors applied to calculate the source contribution. Line 284: why does 0 and 100% represents degradation-dominated and source-dominated, respectively? I think this is conflicted with the main assumption that nitrate transport is dominated by the transit time or flow path (i.e., hydrology-dominated). Besides, Line 284-297: the authors introduced Damköhler number in the method but did not really use it in the rest part of the manuscript. I guess they intended to define transport- and reaction-limited system?

Response #4:

Thanks for pointing that out. What we wanted to express was that the variation of in-stream concentration may attribute to two aspects: (i) the availably of nitrite source changes over time, and (ii) the reaction (degradation) along the flow path. The term "degradation-dominated" may be a more accurate expression than the term "hydrology-dominated", because the degradation is a combination of transit time (influenced by flow paths) and degradation rate (influenced by various factors such as temperature).

However, in the revision, we removed the discussion of source contribution, because the new framework we used for the N source pool does not allow for the calculation of the source contribution. Instead, we only pointed out that 'the $C_Q$ fluctuation is attributed more to the variability in transport rather than to the variability in the N source, echoing previous observations that 80% of the leaching N mass is degraded during transport. However, it is still hard to tell whether the N source or the transport is dominating the $C_Q$ fluctuation' (line 508-511).

For the Damköhler number, yes, we were trying to discuss the transport- and reaction- limited system. However, in the revised version, we simply deleted the Damköhler content as we did not discuss this concept.

5) In-depth discussion about nitrate export is needed. The main goal of this work is to build a connection between hillslope topography and in-stream nitrate concentration through flow paths and water age. However, the current discussion mainly focused on the influence of hillslope topography on subsurface flow paths and water age. Their linkages with nitrate export and how the numerical results here are related to previous field and modelling work are largely missing. Besides, this work is based on **hillslope-scale numerical experiments r**ather than catchment- scale modelling. The authors should also consider the s**cale differences** when they directly applied results

from this work to explain observations at the catchment scale.

Response #5:

We agree. Beside the flow paths and water ages, we added more discussion about the nitrate fluxes between different scenarios, including the N loads (Figure 8a), the leaching, degradation in water and export (figure 8b). Please check section 4.3 for these discussions. We also tried to relate our simulated results with previous studies by adding extra discussions in lines 786-790, 823-838.

Thanks for pointing out the scale issue. We agree that the 'hillslope-scale numerical experiments" may lead to conclusions limited to the specific scale. In the revised version, we updated the experiments using a 3D catchment (Figure 1b). The simulation with the 3D catchment is more suitable to examine the effect of topographic slope compared to a 2D hillslope. We also added 'a 3D model study' in the title.

Other comments with line number:

Line 215-219 and Figure 2: Is the constant source calculated from flow averaged or time-variant concentration averaged? I think it needs to be flow averaged. Otherwise, the overall input mass is not consistent between the two scenarios.

Response #6:

We used a time-weighted average rather than a flow-weighted average. Because we used a constant rainfall over the year, that means the same amount of nitrate was put into the system.
However, in the revised version, we replaced the nitrate source curve with the framework of tracking the N fate in soil (Figure 2). With this framework, the overall input mass to the catchment between scenarios is the same (180 kg/ha/yr).

Figure 5: Horizontal flow and vertical flow: how to calculate the percentage? Even in the saturated zone, the water can still flow horizontally and vertically.

Response #7:

For each cell, we check the velocity to see if flow is more horizontal or more vertical. Then we summed up all the volumes of all cells where velocity is more horizontal (or vertical) and calculate the portion to total aquifer volume. We will clarify that with "flow in 34% of the total aquifer volume is more vertical than horizontal" in line 584.

Figure 5 and 7: what about the spatial distributions of nitrate concentrations? I think adding such figures may help the readers understand the modelling and results.

Response #8:

Thanks for the suggestion, we added figure S2 in the supporting information for the spatial distributions of SIN and DIN (nitrate) loads in the catchment, at the time when SIN and DIN reached their minima and maxima. The figure was cited in the text as "These low and high peaks of SIN and DIN loads can also be identified by the spatial distributions in the catchment (see Figure S2 in the supporting information)" (line 485-487).

Line 355-360 and Figure 5: The flow fluxes into the land surface take a high proportion in the case of slope 1:20 and 1:60. Does nitrate continue to degrade actively in the land surface? If it does, this may not really hold true for real nature systems. Please clarify and discuss the potential implication.

Response #9:

Thanks for pointing that out. The degradation was not considered on the land surface. We will clarify that as "Degradation is not considered on land surface (denitrification in surface flow), where the aerobic condition is more likely to deactivate the denitrification and residence time is short" (line 323-325)

**RC2:**

Overall, this is a nice contribution seeking to explore the impact of the slope of a hillslope transect on nitrate transport and export with the aid of numerical simulations. The analysis is developed using a coupled numerical model for water and solute transport, which also simulates water ages. The topic is of interest to the readership of HESS and I think the Ms could make a good addition to the literature. The text is in general well written and properly organized. However, there are a few limitations that I would like to emphasize in what follows.

I think this is not the first study to explore the impact of the slope of water ages using numerical tools (e.g. Zarlenga and Fiori, 2020), nor the first study to model the export of nitrogen in the context of water ages (van der Velde et al., 2012; Benettin et al., 2020) so I would better put this work in the context of the state of the art. Jasechko et al., 2016 should not be the only conerstone for this study, as it seems to be at times.

Response #10:

Thanks for pointing that out!
We updated the introduction to avoid the impression that our study is the first numerical study to investigate the effect of slope on ages and N export by adding "a 3D model study" in the title and by rephrasing the sentence to "The effect of topographic slope on CQ has rarely been subject to systematical testing" (line 84).

We also included more previous studies in the state of the art, by adding: "Zarlenga et al. [2022] numerically quantified the relative contributions of hillslopes and the drainage network to age dynamics in streamflow, considering the influences of transmissivity and recharge, without focusing on topographic slope" (line 82-84), "For example, Zarlenga and Fiori [2020] presented a physically-based framework to model the transient water ages at the hillslope scale, which was later used to investigate the impact of hillslopes and channels on water ages in catchments [Zarlenga et al., 2022]. Van der Velde et al. [2012] constructed a lumped numerical nitrate transport model for the Hupsel Brook catchment in the Netherlands, without resolving the spatially-explicit details." (line 102-106), "Zarlenga et al., [2022] used a physically-based semi-analytical model to solve the transient water ages in a catchment, without considering surface runoff and hydrological losses (e.g. ET)" (line 123-125), and "…bedrock slope and shape of basin…[ Zarlenga and Fiori, 2020]" (line 855-857).

**The lack of empirical data** to constrain the underlying model parameters is a little bit worrisome. I understand it is difficult to have a full comprehensive analysis of the uncertainty owing to significant computational times, but the authors should put more effort in demonstrating that their simulations are a r**easonable representation of the real world**. I would add more simulations under different scenarios in terms of model parameters, trying to identify how the results obtained in the paper could change if some settings of the numerical simulations are modified (e.g. profile likelihood, sensitivity

analysis). A lot of parameters are simply assumed a priori.

Response #11:

Thanks for the suggestions.
In the revision, we changed the modelling site from the 2D hillslope to a 3D catchment. The measured data for the 3D catchment is used for comparison with the modeled ones, including nitrate concentration measured at the outlet (Figure 4 and Figure S1 in supporting information) and the nitrogen surplus data (line 367-369). The simulated N fluxes and loads in the soil were verified by comparing it to literature values (line 457-474). These showed that our simulated values were fit well and comparable to the observations and that our model was a reasonable representation of the real world.

With regard to parameter sensitivity, in this study we only tried to consider the effect of topographic slope, i.e. the influence that topographic slope has on water ages, N fluxes and concentrations. For other parameters that are likely to change the flow patterns and age compositions (such as catchment size, shape, bedrock topographic and so on) we only discussed them in the limitations section (line 853-863). We plan to consider their effects in future research (line 864-865).

In the revised version, the key parameter values for nitrogen source and transport (Table 2) were either carefully selected according to our available measurements and literature (line 458-490), or optimized with model calibration (Line 333-340).

The way evapotranspiration is treated in the transport model is not described in detail. This is a key process in this context (e.g. changes in the uptake depth of the roots might have a strong impact on the results in some cases) and more emphasis should be given to describe how the numerical code models the green water.

Response #12:

Thanks for the suggestions!
We clarified the computation of ET in the HydroGeoSphere model by adding "ET was simulated as a combination of plant transpiration from the root zone (top 0.5 m soil) and evaporation down to the evaporation depth (0.5 m), which are both constrained by soil water saturation".

We also provided a brief review of the flow model to show how the green water was modeled with HydroGeoSphere (line 207-232).

We also added sentences to explain how the nitrate transport was handled with ET as "To implement the evapoconcentration effect in the transport model, ET is assumed to remove DIN mass without altering the DIN concentration of the water, and to inject that mass back to the SIN pool. This represents a precipitation process from DIN to SIN, which is the inverse process of leaching (Figure 2b). There are two reasons for doing that: (i) the physical process that ET cause the immobilization of DIN can be

mathematically considered, and (ii) N mass balance can be conserved as the plant-uptake is already considered in the N pool according to the plant growth function (Equation 4 and 5), being independent from the ET flux" (line 326-332).

Nitrate is here described as a decaying solute, but I'm not fully convinced by the explanation given by the authors to justify their approach. In particular, I'm not sure that **biogeochemical processes** other than degradation that take place during the transport processes along the hillslope can be completely ignored (i.e. treated as an off-line mechanism that impacy only the initial condition C_J) especially if the solute export is the final goal of the study. More emphasis should be given to the export in the paper as compared to the "transport" issue.

Response #13:

Thanks for pointing that out!
In the original version of the manuscript, we used the approach of an N source concentration curve, which forced the N concentration in the rainfall to change along the curve at the moment of entering the soil. The biogeochemical processes happening in the nitrogen source zone were ignored. This approach did cause a mass imbalance of the system. To overcome this limitation, in the revised version, we used a different framework, which has been used in the study of *Yang et al., 2021*, to simulate the fate of N in the N pool (shallow soil zone), including the external N input, mineralization, degradation in soil, crop-uptake and leaching into groundwater (Line 266-308, Figure 2). This framework increases model complexity by introducing new parameters, however, it also ensures a more reasonable representation of the real-world catchment (see response #11).

Reference:

Yang, J., Heidbüchel, I., Musolff, A., Xie, Y., Lu, C.*, Fleckenstein, J.H. (2021). *Using nitrate as a tracer to constrain age selection preferences in catchments with strong seasonality*, Journal of Hydrology, 603, 126889.

Generalizability issues should be disccused more deeply. How these results might apply to other settings beyond the specific case study presented in the MS and the role of the 3D complexity of a catchment, which is not modeled here? Why do the authors believe their findings are general?

Response #14:

We agree with that!
We made several main updates to the discussion:

- We tried to link our simulated water ages and N fluxes/loads with other previous studies to show that our model is a reasonable representation of the real world catchment under the temperate humid climate (section 4.1, line 457-487). In this way, the conclusion made in this study is more likely to apply to other catchment under similar climate.

- The effect of topographic slope on water ages (and young water fractions) is generally consistent with the findings in previous studies [Bishop et al., 2004; Seibert et al., 2009; Frei et al., 2010; Jasechko et al., 2016] (line 786-792). With the role of the 3D complexity of a catchment (e.g. flow paths in catchment), we were able to identify that the change of the connectivity between the stream and the more distant hillslopes induced the changes of young streamflow fraction (line 792-799).
- We were very careful when describing our conclusion found in the catchment under the temperate humid climate by adding the "a 3d model study" in the title and by discussing the limitations in that it neglects other factors that may also have important impacts on the young streamflow and nitrate export processes. (line 850-851). We emphasized that parameters/settings other than the specific case of this study, such as different landscape aspect, catchment area, aquifer permeability or drainage ability, aquifer depth, stream bed elevation, fractured bedrock permeability, bedrock slope and shape of basin, can potentially change the flow patterns and age composition in streamflow (line 852-857) as well.

Minor points

96: a fairer chain of references here – especially if you talk about TTD of ET - should be Botter et al., 2010; 2011; Van der Velde et al., 2012; Rinaldo et al., 2015, Harman et al., 2015, 2019.

Response #15:
We apologize for missing Botter et al. here for the concept of TTD. We added their work here (line 114) and in the references accordingly.

123-124: this seems to be somewhat speculative at this stage. Move to the   discussion

and elaborate plz.

Response #16:

Thanks! We moved this sentence to the end of the manuscript as "The results of this study improved the understanding of the effects of certain catchment characteristics on nitrate export dynamics with potential implications for the management of stream water quality and agricultural activity" (line 929-931).

150: maybe "Climate" instead of "Climates"?

Response #17:

We changed accordingly to "climate" throughout the text.

165: plz explain in the caption the motivation for the 6 shaded regions represented in panel b of the Figure

Response #18:

We explained the legend for the 10 regions (they became 10 zones as we used a 3D catchment in the revised version) as "Ten aquifer property zones in (b) were defined in the subsurface of the catchment for zonal parameter values (e.g. the hydraulic conductivity)" (line 190 -191).

195-205: plz provide more details about the boundary conditions at the bottom of the domain.

Response #19:

We clarified the boundary conditions for the 3D flow model as "ET was simulated as a combination of plant transpiration from the root zone (top 0.5 m soil) and evaporation down to the evaporation depth (0.5 m), which are both constrained by soil water saturation. Regarding the flow boundary conditions, spatially uniform and temporally variable J was applied to the land surface. Spatially constant and temporally variable potential ET was applied to the aquifer top to calculate the actual ET. The bottom of the aquifer was considered as an impermeable boundary. A critical depth boundary condition was assigned to the catchment outlet to simulate the stream discharge Q, which was compared to the measured Q during the calibration" (line 223-229).

260: not sure this is correct. Plz double check. Why a Deltat is needed from the physical viewpoint? This should be something in the continuous-time domain. Morevoer, the T should appear also on the r.h.s. of the equation (I see there is some text on this in the following lines but I would polish the expression a little bit to make it consistent with the existing literature).

Response #20:

Thanks for pointing that out. We double checked and the $\Delta t$ should be there in equation 9 (line 405). As shown in the following equation, the mass fraction of the tracer $i$ is actually the integration of age distribution over a time step $\Delta t$, over which the tracer $i$ is applied:

$$\int_{t-t_0^i-\Delta t}^{t-t_0^i} p_{Q/ET/S}(T, t) \, dT = \frac{M^i(t)}{\sum M^i(t)}$$

We corrected the expression of water age $T$ as "$T$ is the age ranging within $[t -$

$t_0^i - \Delta t, t - t_0^i]$ for tracer $i$" (line 410).

Figure 4: mean should be intermediate here
Response #21:
Thanks for the suggestion! The Figure 4 of the original manuscript was replaced by the new Figure 6 in the revised version, in which only the flow weighted mean young water fractions were plotted.

Line 430: what about continuous instead of monotonous?
Response #22:
Thanks for the suggestion! We would like to keep using "monotonous" as it may be more precise than "continuous" to state that the response is always rising or falling (line 808-812).

**Editor:**

* The expectations raised at the beginning are nor really kept and should probably be tuned down a bit. One expects a general study on the role of slope on nitrate transport. The manuscript presents a numerical study with a very specific and pretty simplified (as discussed in the end, so this is fine) setup. Although the results are interesting as they reveal different processes leading to different water ages in the stream depending on the slope and the resulting geometrical configuations with surface and groundwater table, they are specific for the setting studied. No attempt is made to generalize the results in terms of scaling to other length scales, extending to 3d, having different land cover etc. So it remains unclear if the effects are general and would also be found in this way in larger (or smaller) catchments.

Response #23:

We agree with that!
We made several main updates to the manuscript:
- We were very careful when describing our conclusion found in the specific catchment under the temperate humid climate by adding the "a 3d model study" in the title and by discussing the limitations in that it neglects other factors that may also have important impacts on the young streamflow and nitrate export processes. (line 850-851). We emphasized that parameters/settings other than the specific case of this study, such as different landscape aspect, catchment area, aquifer permeability or drainage ability, aquifer depth, stream bed elevation, fractured bedrock permeability, bedrock slope and shape of basin, can potentially change the flow patterns and age composition in streamflow (line 852-857) as well.
- We changed the modelling site from a 2D hillslope to a 3D catchment. The measured data for the 3D catchment is used for comparing with the modeled ones, including nitrate concentration measured at the outlet and the nitrogen surplus data. Additionally, we tried to link our simulated water ages and N fluxes/loads with other previous studies to show that our model is a

reasonable representation of the real world catchment under the temperate humid climate (section 4.1, line 457-487). In this way, the conclusion made in this study is more likely to apply to other catchment under similar climate.

* Although the simplifications are pointed out at the end of the paper, the discussion on their relevance for the results is rather short. In particular the 2d setting raises questions and also the boundary conditions chosen in the model. The transport is modeled in a simple way and it is unclear what assumptions were made concerning the mixing (which plays an important role for the age distribution). To learn something more general, a bit more discussion would be useful.

Response #24:

Thanks for pointing out that!

In the revised version, we updated the modelling from a 2D hillslope to a 3D catchment. The measured data for the 3D catchment is used for comparing with the modeled ones, including nitrate concentration measured at the outlet and the nitrogen surplus data. The measured data for the 3D catchment is used to compare with the modeled ones, including nitrate concentration measured at the outlet (Figure 4 and Figure S1 in supporting information) and the nitrogen surplus data (line 367-369). The simulated N fluxes and loads in soil were verified by comparing with literature values (line 457-474). These demonstrated that that our simulated values were fit well or comparable to the observations and our model is a reasonable representation of the real world. We clarified the boundary conditions for the 3D flow model (line 223-229).

For the N transport, we employed a different framework to simulate the nitrogen fluxes in the soil, including the input, mineralization, degradation in soil, crop-uptake and leaching into groundwater. This framework had increased model complexity by introducing new parameters, however, it ensured a correct mass balance and delivered nitrogen fluxes in the soil.

Concerning the mixing of groundwater that influences the age distributions, we added discussion as "…Because of the three aforementioned effects, the connectivity between the stream and the more distant hillslopes is significantly reduced. Precipitation falling farther from the stream has a lower chance to reach the stream and a higher change to be intercepted by ET on its way to the stream. The hillslope that used to generate old streamflow does not contribute to streamflow anymore…" (line 635-638), and "the reduction of connectivity, which results in the reduction of the longer flow paths and the persistence of shorter flow paths, causes the increase of the young streamflow fraction…" (line 797-799).

For the results part, more discussion was added for the effect of topographic slope on nitrogen fluxes, including leaching, degradation, and export to the stream (line 684-698), and in a more general way as "Our results demonstrate that stream water quality is potentially less vulnerable in flatter landscapes. The flatter landscapes tend to retain more N mass in the soil and export less N mass to the stream…" (line 816-829).

* The third 'class' (and I would be careful with such general terms if the generalization of the results is not made), where the setting is very flat, is described in the way that infiltrating water does not reach the stream by lateral transport, as it goes into ET before it reaches the stream. It would be useful to illustrate for all cases the contributions of the different parts in the model to the different parts in the water budget. If I understand correctly, ET is the same for all cases. Also the size of the domain is the same for all cases. So the same amount of water goes into ET for steep and flat cases. This implies that for steep settings the water that goes into ET has in total to come from deeper parts and has to be 'older' than for flat settings. Or is ET itself just higher for flat settings because the soil is wetter? These things do not get very clear.

Response #25:

Thanks for pointing out that!
We removed the term "class" from our discussion.

We updated that discussion about the effect of topographic slope on groundwater flow by clarifying the changes of Q, ET, groundwater table, and young water fraction with different settings (line 553-568, Figure 6).

For the 3D catchment, the simulation result showed that the ET is high for flatter landscapes, rather than being constant in different settings. We clarified that as "Figure 6 shows the responses of temporally-averaged Q and ET, the groundwater table depth, and flow weighted mean $YF_Q$ and $YF_{ET}$ to the changes of topographic slope. Under a constant climate, the changes of topographic slope can reshape the water flow via influencing flow partitioning between Q and ET. More water is taken up by ET and less water becomes Q in flatter landscapes (Figure 6a). These patterns can be explained by the change of groundwater table depth (Figure 6b), as shallower groundwater tables can be reached by the vegetation in flatter landscapes where ET therefore has a higher chance to remove water from the subsurface." (line 554-562).

The mechanical explanation for the response of the young water fractions to the slope was updated in the revised version as "This transformation leads to three main effects: …" (line 607-634) and "Because of the three aforementioned effects, the connectivity between the stream and the more distant hillslopes is significantly reduced…" (line 635-642).

---

## Author Response (AR2)

Dear reviewers and editor,

please check our responses to your constructive comments (marked in blue). Unless specified, all the line numbers in our responses refer to the line numbers of the TRACK CHANGE version of the manuscript. Thank you!

Response overview:

We have made updates to the manuscript, main updates are: (i) more details about calibration were added, and (ii) updating the introduction by including more literatures. We would like to acknowledge the efforts from the editor and the reviewers for the MS.

**RC1:**

The main goal of this paper, a resubmission of a manuscript I have already reviewed, is to investigate the impacts of topographic slope on subsurface flow, water age, and nitrate export at the catchment scale. Compared to the previous version, the paper has been significantly improved and all my major comments have been satisfactorily addressed. One minor request that I would make is to list the specific ranges for those zonal values in Table 1.

Response #1:

Thanks for the positive comment. We updated the Table 1 by including the two ranges for the zonal hydraulic conductivity and porosity values, respectively (**line 223**).

**RC2:**

I have appreciated the changes made by the authors to the Ms. in response to the comments of the referees. I think the paper is now improved as compared to the original submission, and the scope is certainly broader owing to the addition of observational data. However, the broader scope implies more issues to face. In particular, I suggest to provide more context / justification / details about the calibration procedure - the range of values for each model parameter explored (prior distribution), the possible definitions of the objective functions (e.g. using only Q data for the calibration of the hydrologic params, and then use C data for reactive transport params vs. all params being calibrated at once with a global obj function related to C and Q), the choice of fixed params vs calibrated params, the ensuing posterior distributions, the potential uncertainty. While I understand a full uncertainty analysis could be unfeasible in this case, the impact of operational choices done by the authors in their calibration exercise need to be better assessed/discussed. Moreover, the literature about nitrogen modeling in soils is huge, and more refs could be added to the Ms. Overall I contgratulate the authors for their efforts.

Response #2:
Thanks for the suggestions. We added more details to the calibration procedure, and

discussed several impacts of the calibration exercise, accordingly. The main updates are:

- We clarified how the objective function in this study was defined using the data sets for transport as "PEST uses the Marquardt method [*Marquardt*, 1963] to minimize a target function by varying the values of a given set of parameters until the optimization criterion is reached. We used the measured $C_Q$ and N surplus as the target variables for comparison with the simulated ones. The N surplus, which is the annual amount of N remaining in the soil after consumption by plant-uptake, was estimated as 48.8 kg ha$^{-1}$ yr$^{-1}$ [*Yang et al.*, 2021). As two different data sets ($C_Q$ and N Surplus) were used, a weighting scheme was used such that the defined multi-objective function was not dominated by one data set" (**lines 304 – 309**).

- We clarified that our calibration actually followed a procedure of two steps: first for flow, and second for transport. The potential effect of choosing this two-steps procedure instead of calibration all at once was also mentioned and discussed. These clarification was done by adding "Note that the entire model calibration (for flow and transport) actually followed a procedure of two steps: first for flow, and second for transport. Alternatively, the flow and transport parameters can be calibrated at one step by defining the multi-objective function using all the data sets (discharge, groundwater levels, CQ and N surplus). The potential effect of the two different calibration procedures on the modeling results should be further explored, however, being out of the main focus of this study. We consider the two-step calibration procedure to be acceptable, because our result showed that it was sufficient to reach an acceptable model performance for both flow and transport (described later)" (**lines 310 -316**). We also added "As the flow parameters (e.g., hydraulic conductivity and porosity) were already calibrated in Yang et al. [2018] using data sets of discharge and groundwater levels. In this study, the calibration was only performed for the transport…" (**lines 301-302**) at the beginning such that the readers can be clearer with the calibrations.

- We clarified that several parameters were fixed and others were adjustable for calibration by adding "Several transport parameters were fixed at the values selected according to prior information, such that the degree of freedom in the calibration can be reduced as much as possible (Table 2). In total eight parameters were adjustable and calibrated, because they were the key parameters to determine the N fluxes in soil and groundwater" (**lines 317-319**).

- The ranges of the adjustable model calibrations were listed in table 2 and described in text as "Their adjustable ranges were selected according to the literature or to cover the values that the parameters can realistically reach (Table 2)" (**lines 319 -321**).

- We emphasized the potential model uncertainty in the simulated N loads and fluxes in the discussion section 4.5 as "While the numerical model provided general insights, there was potential uncertainty in the simulated results. Firstly, the aforementioned simplifications may introduce model structural errors. Secondly, the model calibration was only constrained by limited data sets, which may lead to the non-uniqueness in the model parameters. Both of the aspects may introduce uncertainty in the simulated N loads and fluxes. Future work should be devoted to better constrain the model parameters, either by enhancing the concentration data quality through more frequent measurements or by providing additional data sets related to the N pool" (**lines 628-633**).

Response #3:

Thanks for the pointing that out. We added more literatures studding the nitrogen dynamics in catchments to the introduction, as "A number studies focused on numerically simulating the nitrogen fluxes (or loads) in soil and groundwater [*Smith et al.,* 2004; *Rivett et al.,* 2008; *Lindström et al.,* 2010*; van der Velde et al.,* 2012; *Van Meter et al.,* 2017; *X. Yang et al.,* 2018, 2019; *Kolbe et al.,* 2019; *Knoll et al.,* 2020; *Nguyen et al.,* 2021, 2022].  For example, *van der Velde et al.* [2012] constructed a lumped numerical nitrate transport model for the Hupsel Brook catchment in the Netherlands. *Lindström et al.* [2010] developed HYPE water quality model allowing for simulating the nitrogen fluxes in soil. *Van Meter et al.* [2017] investigated the two-centuries nitrogen dynamics in the Mississippi and Susquehanna River Basins using a TTD (transient time distribution) based transport approach. *X. Yang et al.* [2018] developed the coupled mHM-Nitrate model, which can provide valuable insights int the spatial variability of water and nitrate fluxes in catchment scale. *Nguyen et al.* [2021] further updated that model to the mHM-SAS model by implementing the SAS-function based solute transport module [*Harman*, 2015, 2019; *Rinaldo et al.*, 2015; v*an der Velde et a*l., 2012], allowing for simulating the nitrate export from a Mesoscale Catchment. However, most of these works provided little information on the spatially-explicit details (such as the flow field) for interpreting the nitrate dynamics" (**lines 91 -104**).

The reference list was updated accordingly.

---

## Author Response (AR3)

Dear Prof. Neuweiler,

Thank you for the comments for our last submission. The minor technical correction was made accordingly (line 302). Beside the minor technical correction, we would like to further clarify the issue of "prior distribution" and "posterior distribution" here (not in the MS as I noticed that I am not allowed to make any more changes to the accepted manuscript except the changes requested by the Editor).

The software PEST actually uses the Gauss-Marquardt-Levenberg (GML) method [*Marquardt*, 1963; *Draper and Smith*, 1998] to minimize a target function by varying the values of a given set of parameters until the optimization criterion is reached. GML method, as a frequentist approach, considers the unknown parameters as fixed and attempts to estimate them through the calibration process using the data sets. The outcome of such calibration approach is the most likely estimate of the parameters (best-fit parameters) associated with different levels of confidence. We also noticed that this approach differs from the Bayesian approach (e.g., Markov Chain Monte Carlo method), which considers the parameters as random with the prior distributions and attempts to update the prior ones into the posterior distributions by the conditioning effect of the data sets. The Bayesian approach usually has higher computational cost and its performance can be hampered for complex models. Therefore, it was not suitable for our model with relatively long CPU time (4 hours per run) and involved with various flow and transport process.

Please let us know if you have any further questions regarding the updates to the manuscript. Thank you!

Best regards
Jie